# Causal Representation Learning with Optimal Compression under Complex Treatments

**Wanting Liang** [1]   **Haoang Chi** [3]   **Zhiheng Zhang**[⊠ 1 2]

## Abstract

Estimating Individual Treatment Effects (ITE) in multi-treatment scenarios presents two challenges: the hyperparameter selection dilemma and the curse of dimensionality. We derive a novel generalization bound and propose a theoretically grounded estimator for the optimal balancing weight $\alpha$, eliminating expensive heuristic tuning. We investigate three balancing strategies: Pairwise, One-vs-All (OVA), and Treatment Aggregation. While OVA excels in low-dimensional settings, our Treatment Aggregation strategy ensures $\mathcal{O}(1)$ scalability. Furthermore, we extend this framework to Multi-Treatment CausalEGM, a generative architecture preserving the Wasserstein geodesic structure of the treatment manifold. Experiments on semi-synthetic and image datasets demonstrate that our approach significantly outperforms traditional models in estimation accuracy and efficiency, particularly in large-scale intervention scenarios.

## 1. Introduction

Estimating individualized causal effects from observational data is a central problem in modern causal inference, with direct implications for personalized medicine, policy evaluation, and targeted interventions. Recent progress has been driven by *causal representation learning* (Schölkopf et al., 2021), which learns a latent representation $\Phi(X)$ in which treatment assignment becomes less confounded while outcome-relevant information is retained (Johansson et al., 2016; Shalit et al., 2017). Despite its empirical success, the statistical role of representation learning remains in-

[1]School of Statistics and Data Science, Shanghai University of Finance and Economics, Shanghai, China [2]Institute of Big Data Research, Shanghai University of Finance and Economics, Shanghai, China [3]National University of Defense Technology, Changsha, China. Correspondence to: Zhiheng Zhang <zhangzhiheng@mail.shufe.edu.cn>.

*Proceedings of the 43rd International Conference on Machine Learning*, Seoul, South Korea. PMLR 306, 2026. Copyright 2026 by the author(s).

completely understood beyond the binary-treatment regime. Recent work has begun to open the representation layer itself, providing a more fine-grained characterization of latent structure and its interaction with high-dimensional covariates (Liu et al., 2024). However, this focuses on covariate-side complexity, and leaves open how representation learning behaves when the treatment space itself is multi-dimensional or endowed with richer geometric structure, as in dose–response or geodesic causal settings (Kurisu et al., 2024).

A key methodological distinction of causal representation learning—relative to weighting- or matching-based paradigms—is that it enforces *invariance through compression*. Concretely, many objectives take the form $\mathcal{L}(\theta; \alpha) = \mathcal{L}_{\mathrm{pred}}(\theta) + \alpha\,\mathcal{R}_{\mathrm{bal}}(\theta)$, where $\mathcal{R}_{\mathrm{bal}}$ measures distributional discrepancy across treatment groups (often via an IPM), and $\alpha$ controls the fundamental trade-off between *confounding removal* and *information preservation*. Prior binary-treatment theory implies that an optimal $\alpha^\star$ exists (Shalit et al., 2017), yet in practice $\alpha$ is almost always treated as a heuristic hyperparameter.

This heuristic breaks down sharply in *multi-treatment* settings ($T \in \{0, \ldots, K-1\}$), where interventions naturally come in many levels (e.g., drug dosage, regimen choice, or multi-channel marketing). Consider personalized dose-response estimation with tens of dosage levels and high-dimensional covariates (images, text, EHR): choosing $\alpha$ by grid search becomes prohibitively expensive, while confounding patterns vary across treatment arms, making tuning unstable. Moreover, prevailing multi-treatment extensions typically enforce *pairwise* balancing (Schwab et al., 2018; Lopez & Gutman, 2017), requiring $\mathcal{O}(K^2)$ discrepancy constraints; this not only makes training unscalable, but can also *over-constrain* $\Phi$ and induce representation collapse, deteriorating both predictive accuracy and causal generalization. Crucially, this is *not* simply a matter of summing more binary-style terms: all $\binom{K}{2}$ pairwise constraints are optimized through the *same* shared map $\Phi$, forming a statistically *coupled* system whose behavior under growing $K$ has no binary-treatment analogue and has remained entirely uncharacterized.

These observations expose a missing theoretical ingredient.

Recent theoretical evidence already suggests that causal representation learning does not admit a "free lunch": compressing representations to reduce imbalance between treatment and control groups can itself introduce additional bias in identification and estimation (Melnychuk et al., 2023). This phenomenon highlights that invariance is not uniformly beneficial, but comes at a statistical cost that depends on how aggressively the representation is compressed. Current literature lacks a principled characterization of *how much invariance a causal representation should enforce* when $K$ grows: the prevailing—and as we show, *incorrect*—intuition is that stronger, finer-grained balancing is uniformly better, yet the multi-treatment coupled structure amplifies the statistical cost of over-compression in ways that are both theoretically novel and practically consequential. Thus, the core unresolved question is not merely computational—it is statistical: *what is the optimal representation trade-off, and how should $\alpha$ (and the balancing strategy) be chosen in a way that remains stable and scalable under many treatments?*

We address this gap by reframing multi-treatment causal representation learning as a problem of **optimal compression** (see Appendix A for a detailed discussion of related work).

**A key finding: in multi-treatment regimes, stronger balancing is not necessarily better.** This is not merely a computational observation—it reveals a fundamentally new *coupled compression* phenomenon that does not arise in binary settings. When $K > 2$, pairwise balancing introduces $\binom{K}{2}$ discrepancy terms that are statistically coupled through the shared representation $\Phi$. This coupling causes the tuning signal to become unstable as $K$ grows: the deviation rate of $\widehat{\alpha}$ under pairwise balancing scales as $\mathcal{O}(K^2/\sqrt{n})$, and under mild cross-arm dependence its variance scales as $\Theta(K^4/n)$—orders of magnitude worse than the binary case. In contrast, our Treatment Aggregation strategy controls imbalance through a single global functional (HSIC), achieving $K$-independent stability $\Theta(1/n)$. This theoretical finding changes the practical takeaway: beyond a certain scale, enforcing more constraints can *over-constrain* the representation, destroy prognostic signal, and harm causal generalization—making scalable, coarse-grained balancing strictly preferable to fine-grained pairwise alignment.

In summary, our results provide a unified answer to the central question of multi-treatment causal representation learning: *how to choose an optimal representation that balances deconfounding and information preservation, without heuristic tuning and without quadratic scaling in $K$.* Our main contributions are: (1) We derive a multi-treatment generalization bound that formalizes the bias–information trade-off and yields a consistent estimator for the optimal balancing weight $\alpha^\star$ (eliminating expensive heuristic tun-

ing). (2) We propose Treatment Aggregation via treatment embeddings and HSIC, achieving $\mathcal{O}(1)$ balancing complexity and stable behavior as the number of treatments increases. (3) We extend the framework to a generative architecture and demonstrate Wasserstein-geodesic consistency, enabling physically interpretable counterfactual interpolation across treatments.

## 2. Framework

We observe i.i.d. samples $\{(X_i, T_i, Y_i)\}_{i=1}^n$ from a distribution on $\mathcal{X} \times \mathcal{T} \times \mathcal{Y}$, with covariates $X \in \mathcal{X}$, outcome $Y \in \mathcal{Y}$, and discrete treatment $T \in \{0, \dots, K-1\}$. Let $P_{Y(t)|X=x}$ denote the conditional distribution of potential outcomes $\{Y(t)\}_{t \in \mathcal{T}}$, assuming standard identification conditions (Assumption 2.4).

Throughout, we treat individualized causal effects *distributionally* in Wasserstein space. For any $j, k \in \mathcal{T}$, define

$$\tau_{j,k}(x) := \mathcal{W}_2\big(P_{Y(j)|X=x}, P_{Y(k)|X=x}\big), \qquad (1)$$

where $\mathcal{W}_2$ is the 2-Wasserstein distance (well-defined whenever the involved distributions have finite second moments). A natural target risk is the multi-treatment **Precision in Estimation of Heterogeneous Effects** (PEHE) criterion

$$\epsilon_{\mathrm{ITE}} := \mathbb{E}_X\left[\sum_{0 \leq j < k \leq K-1} (\widehat{\tau}_{j,k}(X) - \tau_{j,k}(X))^2\right], \quad (2)$$

where $\widehat{\tau}_{j,k}(x)$ is computed from the model's estimated conditional distributions. In the classical scalar-outcome regime where one models conditional means (or for degenerate $P_{Y(t)|X=x}$), (1) reduces to a mean-based ITE (up to embedding), hence (2) subsumes the standard PEHE objective studied in Johansson et al. (2016); Shalit et al. (2017).

We learn a representation map $\Phi : \mathcal{X} \to \mathcal{Z} \subset \mathbb{R}^d$ and a predictor $h : \mathcal{Z} \times \mathcal{T} \to \mathcal{P}_2(\mathcal{Y})$, where $\mathcal{P}_2(\mathcal{Y})$ denotes distributions on $\mathcal{Y}$ with finite second moments. Given $z = \Phi(x)$, the model outputs an estimated conditional distribution $\widehat{P}_{Y|z,t} := h(z, t)$, hence $\widehat{\tau}_{j,k}(x) := \mathcal{W}_2\big(\widehat{P}_{Y|\Phi(x),j}, \widehat{P}_{Y|\Phi(x),k}\big)$. Let $\ell(\widehat{P}, y)$ be a proper loss for distribution prediction (e.g., negative log-likelihood for a parametric family, or a Wasserstein-compatible surrogate). Define the factual risk

$$\begin{aligned} \epsilon_F(\Phi, h) &:= \mathbb{E}\big[\ell(h(\Phi(X), T), Y)\big], \\ \widehat{\epsilon}_F(\Phi, h) &:= \frac{1}{n} \sum_{i=1}^n \ell(h(\Phi(X_i), T_i), Y_i). \end{aligned} \qquad (3)$$

We now recall the key structural insight from the binary-treatment literature ($K = 2$). When $h$ parameterizes only conditional means (degenerate outcome distributions), $\ell$

reduces to squared loss and $\epsilon_{\mathrm{ITE}}$ reduces to the classical PEHE of Shalit et al. (2017).

In Johansson et al. (2016), the counterfactual risk is bounded via a domain-adaptation argument: the discrepancy between treated and control domains enters through an IPM in representation space. Building on this, Shalit et al. (2017) derives binary-treatment generalization bounds for ITE estimation (PEHE) of the schematic form $\epsilon_{\mathrm{ITE}}(\Phi, h) \leq C_F \epsilon_F(\Phi, h) + C_B \mathrm{IPM}_{\mathcal{G}}(P_{\Phi}^{(0)}, P_{\Phi}^{(1)}) + C_C \mathrm{Complexity}(h \circ \Phi; n, \delta)$ for suitable constants and regularity conditions (e.g., Lipschitz hypotheses and bounded losses). The crucial point is the *decomposition*: ITE error is controlled by (i) factual prediction and (ii) a representation-level imbalance term. This decomposition motivates posing representation learning as a *controlled compression* problem. The positive constants $C_F, C_B, C_C$ depend on regularity conditions (e.g., Lipschitz/smoothness and boundedness assumptions) independent of $n$, matching specifications in prior work (Csillag et al., 2024; Johansson et al., 2022; Shalit et al., 2017; Johansson et al., 2016).

*Remark* 2.1. For this inequality, in the binary case, $\mathcal{T} = \{0, 1\}$ and $P_{\Phi}^{(t)}$ denotes the conditional distribution of the learned representation $\Phi(X)$ given treatment $T = t$, i.e., $P_{\Phi}^{(t)} := \mathcal{L}(\Phi(X) \mid T = t)$. The *factual risk* is $\epsilon_F(\Phi, h) := \mathbb{E}\big[\ell(h(\Phi(X), T), Y)\big]$, where $\ell(\widehat{P}, y)$ is a proper loss for predicting the (possibly distributional) outcome given $(\Phi(X), T)$ and the expectation is taken over the observational distribution of $(X, T, Y)$. The *ITE risk* $\epsilon_{\mathrm{ITE}}(\Phi, h)$ denotes the target error for individualized treatment effects (e.g., PEHE in the scalar-mean setting), and in our distributional formulation it is the risk in (2) specialized to $K = 2$. The discrepancy term used in the bound is an integral probability metric (IPM): $\mathrm{IPM}_{\mathcal{G}}(P, Q) := \sup_{g \in \mathcal{G}} \big| \mathbb{E}_P[g] - \mathbb{E}_Q[g] \big|$, where $\mathcal{G}$ is a prescribed function class (e.g., the unit ball of an RKHS, yielding MMD). The bound holds with probability at least $1 - \delta$ over the draw of the sample of size $n$, where $\delta \in (0, 1)$ is a confidence level. Finally, $\mathrm{Complexity}(h \circ \Phi; n, \delta)$ denotes a standard statistical complexity term controlling the generalization gap of the composed predictor $h \circ \Phi$ (e.g., via Rademacher complexity or VC-type bounds); its precise form and constants depend on the hypothesis class and regularity assumptions, and will be made explicit in Section 3.

For a balancing strategy $\mathcal{S}$, define a population imbalance functional $\mathcal{R}_{\mathcal{S}}(\Phi) := \mathcal{D}_{\mathcal{S}}\big(\{P_{\Phi}^{(t)}\}_{t \in \mathcal{T}}\big)$, where $\mathcal{D}_{\mathcal{S}}$ is a strategy-specific discrepancy operator (e.g., an IPM-based sum or a dependence functional). Since $\mathcal{R}_{\mathcal{S}}(\Phi)$ is unknown, we use an empirical estimator $\widehat{\mathcal{R}}_{\mathcal{S}}(\Phi)$ computed from $\{(X_i, T_i)\}_{i=1}^{n}$. We interpret $\rho \geq 0$ as an *invariance (balance) budget*. Motivated by the binary bounds above, we define the representation selection problem as: find the most predictive representation among those that satisfy a prescribed imbalance budget,

$$\min_{\Phi, h} \; \widehat{\epsilon}_F(\Phi, h) \quad \text{s.t.} \quad \widehat{\mathcal{R}}_{\mathcal{S}}(\Phi) \leq \rho, \tag{4}$$

which is precisely a *controlled compression* formulation.

Crucially, (4) is (under standard constraint qualifications) equivalent to the *penalized* Lagrangian form

$$\min_{\Phi, h} \; \widehat{\epsilon}_F(\Phi, h) + \alpha \, \widehat{\mathcal{R}}_{\mathcal{S}}(\Phi), \qquad \alpha \geq 0, \tag{5}$$

where $\alpha$ is the Lagrange multiplier corresponding to the budget $\rho$. Thus, tuning $\alpha$ is theoretically equivalent to selecting a feasible representation family $\{\Phi : \widehat{\mathcal{R}}_{\mathcal{S}}(\Phi) \leq \rho\}$, i.e., controlling the effective geometry and capacity of $\Phi$.

**Lemma 2.2** (Penalty–constraint equivalence). *Let $f(\Phi, h) := \widehat{\epsilon}_F(\Phi, h)$ and $g(\Phi) := \widehat{\mathcal{R}}_{\mathcal{S}}(\Phi)$. Consider the constrained problem* (4) *with budget $\rho$. Assume (i) the feasible set $\{(\Phi, h) : g(\Phi) \leq \rho\}$ is nonempty and admits a strictly feasible point (Slater condition), and (ii) strong duality holds for* (4). *Then there exists $\alpha^{\star} \geq 0$ such that every primal optimum of* (4) *is also a minimizer of* (5) *with $\alpha = \alpha^{\star}$. Conversely, for any $\alpha \geq 0$, any minimizer of* (5) *is a minimizer of* (4) *with $\rho = g(\Phi_{\alpha})$, where $\Phi_{\alpha}$ is an optimizer under $\alpha$.*

If $\alpha$ is too small (or $\rho$ too large), the optimizer can choose representations close to the identity map, preserving outcome signal but also preserving imbalance, yielding biased counterfactual generalization. If $\alpha$ is too large (or $\rho$ too small), the optimizer is forced to compress aggressively, often discarding outcome-relevant information. We aim to locate the *optimal compression point* (the optimal $\alpha^{\star}$).

**Why do we treat $\alpha$ as a free (and estimable) multiplier?** A potential confusion is that in the binary bound of Shalit et al. (2017), the imbalance term appears with a constant coefficient, suggesting a fixed ratio (*i.e.*, choose $\alpha = \frac{C_B}{C_F}$) between factual risk and discrepancy. However, this ratio is not an identifiable constant: it aggregates regularity parameters (e.g., Lipschitz/smoothness constants of the hypothesis class and the choice of IPM function class), and it changes under rescaling of losses, representation constraints, and metric choices. We therefore parameterize this unknown ratio by $\alpha$ and interpret representation learning as *controlled compression*. Under standard constraint qualifications, tuning $\alpha$ in the penalized objective is equivalent to selecting a feasible representation family under an imbalance budget, i.e., moving along the Pareto frontier between predictive fidelity and distributional invariance. This perspective is increasingly essential in multi-treatment problems, where the discrepancy structure depends on the balancing strategy and the number of treatments, rendering any fixed-coefficient approach conceptually and practically untenable.

**Extension: From Binary to Multi-Treatment.** When $K > 2$, the binary template immediately raises the

question: *what replaces the single treated–control discrepancy?* A direct extension of the arguments in Johansson et al. (2016); Shalit et al. (2017) treats each pair of treatment arms as a domain-adaptation subproblem. This yields a multi-treatment bound of the same schematic form, $\epsilon_{\mathrm{ITE}}(\Phi, h) \lesssim \widehat{\epsilon}_F(\Phi, h) + \sum_{0 \le j < k \le K-1} \mathrm{IPM}_{\mathcal{G}}\big(\widehat{P}_\Phi^{(j)}, \widehat{P}_\Phi^{(k)}\big) + (\text{complexity term})$. The key difference is that the imbalance component now involves *multiple* discrepancies across arms, which is the source of both computational and statistical instability as $K$ grows. Our theory section (Section 3) will formalize this extension and quantify how different imbalance constructions affect stability.

Let $P_\Phi^{(t)}$ denote the conditional distribution of $\Phi(X)$ given $T = t$. Recall the definition of IPM, we study three strategies $\mathcal{S} \in \{\mathrm{pair}, \mathrm{ova}, \mathrm{agg}\}$:

**Pairwise balancing ($\mathcal{S} = \mathrm{pair}$).**

$$\mathcal{R}_{\mathrm{pair}}(\Phi) := \sum_{0 \le j < k \le K-1} \mathrm{IPM}_{\mathcal{G}}\left(P_\Phi^{(j)}, P_\Phi^{(k)}\right), \text{cost } \mathcal{O}(K^2).$$

**One-vs-All balancing ($\mathcal{S} = \mathrm{ova}$).** Let $P_\Phi^{(-k)}$ be the mixture distribution of $\Phi(X)$ over $\{t \ne k\}$.

$$\mathcal{R}_{\mathrm{ova}}(\Phi) := \sum_{k=0}^{K-1} \mathrm{IPM}_{\mathcal{G}}\left(P_\Phi^{(k)}, P_\Phi^{(-k)}\right), \qquad \text{cost } \mathcal{O}(K).$$

**Treatment aggregation ($\mathcal{S} = \mathrm{agg}$).** We embed treatments via a learnable map $e : \mathcal{T} \to \mathbb{R}^{d_e}$ and write $E_T := e(T)$. We enforce global independence between $\Phi(X)$ and $E_T$ through the **Hilbert-Schmidt Independence Criterion** (HSIC) (Gretton et al., 2005):

$$\mathcal{R}_{\mathrm{agg}}(\Phi) := \mathrm{HSIC}\big(\Phi(X), E_T\big), \qquad \text{cost } \mathcal{O}(1) \text{ w.r.t. } K.$$

For IPM instantiations such as **Maximum Mean Discrepancy** (MMD), $\widehat{\mathcal{R}}_{\mathrm{pair}}(\Phi)$ and $\widehat{\mathcal{R}}_{\mathrm{ova}}(\Phi)$ are obtained by replacing expectations in the IPM definition with sample averages, yielding standard (unbiased) U-statistic estimators. For HSIC, $\widehat{\mathcal{R}}_{\mathrm{agg}}(\Phi)$ is the usual empirical V-statistic estimator computed from $\{(\Phi(X_i), e(T_i))\}_{i=1}^n$. These are consistent estimators of their population counterparts under mild moment conditions (See detailed definitions in Appendix B.1).

**The Optimal Trade-off as a Bilevel Problem.** The population-optimal tuning parameter is

$$\alpha^\star(\mathcal{S}) \in \arg \min_{\alpha \ge 0} \epsilon_{\mathrm{ITE}}\left(\Phi_{\alpha,\mathcal{S}}^\star, h_{\alpha,\mathcal{S}}^\star\right), \qquad (6)$$

where $(\Phi_{\alpha,\mathcal{S}}^\star, h_{\alpha,\mathcal{S}}^\star)$ denotes the optimizer of the population analogue of (5) under strategy $\mathcal{S}$. Since $\epsilon_{\mathrm{ITE}}$ involves counterfactual distributions, it is not directly observable. Instead,

we select $\alpha$ by minimizing an explicit generalization upper bound on $\epsilon_{\mathrm{ITE}}$ derived in Section 3. Operationally, this yields the bilevel procedure:

$$\widehat{\theta}(\alpha, \mathcal{S}) \in \arg \min_\theta \widehat{\epsilon}_F(\theta) + \alpha \widehat{\mathcal{R}}_{\mathcal{S}}(\theta),$$
$$\widehat{\alpha}(\mathcal{S}) \in \arg \min_{\alpha \ge 0} \widehat{\mathcal{B}}_{\mathcal{S}}(\widehat{\theta}(\alpha, \mathcal{S}), \alpha), \qquad (7)$$
$$\widehat{\mathcal{B}}_{\mathcal{S}}(\theta, \alpha) := \widehat{\epsilon}_F(\theta) + \alpha \widehat{\mathcal{R}}_{\mathcal{S}}(\theta) + \mathrm{Comp}_{\mathcal{S}}(\alpha; n, \delta)$$

where $\theta$ collects all parameters of representation $\Phi$ and predictor $h$, and $\widehat{\mathcal{B}}_{\mathcal{S}}$ is the empirical counterpart of the bound.

Here $\widehat{\mathcal{B}}_{\mathcal{S}}(\theta, \alpha)$ denotes an explicit empirical upper bound on the ITE risk $\epsilon_{\mathrm{ITE}}(\Phi, h)$ obtained by replacing population quantities in the generalization bound with their empirical counterparts. Moreover, $\mathrm{Comp}_{\mathcal{S}}(\alpha; n, \delta)$ is a uniform generalization term controlling the gap between population and empirical risks over the $\alpha$-*induced hypothesis class* (see Appendix B.2 for the explicit Rademacher-based definition). Importantly, stronger compression (larger $\alpha$) shrinks the effective class and typically reduces $\mathrm{Comp}_{\mathcal{S}}$, yielding a non-trivial trade-off in $\alpha$. *Remark.* If the remainder term were chosen independent of $\alpha$, then minimizing the bound after optimizing $\theta$ would degenerate to $\widehat{\alpha} = 0$, since the upper-level objective would reduce to the training criterion. Our bound therefore includes an $\alpha$-dependent complexity term, reflecting the fact that stronger compression shrinks the effective hypothesis class and improves generalization.

**On the $n$-dependence of the complexity term.** A natural concern is that the remainder of complexity $\mathrm{Comp}_{\mathcal{S}}(\alpha; n, \delta)$ typically scales as $\mathcal{O}(n^{-1/2})$ and thus vanishes asymptotically. This does not undermine the role of $\alpha$. First, $\alpha$ is selected for *finite-sample* generalization: when $n$ is moderate and the effective model class is rich (especially under many treatments), the $\mathcal{O}(n^{-1/2})$ term can dominate the risk bound and critically determines the optimal compression level. Second, our target is the sample-size dependent minimizer $\alpha_{\mathcal{S}}^\star(n) \in \arg \min_{\alpha \in \mathcal{A}} \mathcal{B}_{\mathcal{S}}(\alpha; n)$, and we establish consistency of $\widehat{\alpha}_{\mathcal{S}}$ for $\alpha_{\mathcal{S}}^\star(n)$. Whether $\alpha_{\mathcal{S}}^\star(n)$ converges to 0 or to a positive limit as $n \to \infty$ depends on the intrinsic bias–information structure of the problem and is not pathological. Finally, in multi-treatment regimes the effective constants in $\mathrm{Comp}_{\mathcal{S}}$ can scale with $K$ in a strategy-dependent manner, making the finite-sample trade-off non-negligible even for large $n$.

*Remark* 2.3 (Geometric Extension: Wasserstein Manifolds and Geodesic Causal Inference). The Wasserstein formulation (1) places potential outcome distributions on a non-Euclidean metric space. In many applications (e.g., dose-response, intervention intensity, biological trajectories), treatments appear discrete but are generated by an underlying continuous mechanism. Geodesic Causal Inference (GCI) (Kurisu et al., 2024) formalizes this viewpoint by modeling causal variation as geodesic flows on Wasserstein

manifolds. In our framework, the *same blueprint* persists: (i) start from a generalization bound whose leading terms decompose into a factual prediction component and a representation imbalance component, (ii) pose a controlled compression problem of the form (4)–(5), and (iii) validate geometric consistency by checking whether counterfactual interpolation follows Wasserstein geodesics rather than Euclidean mixtures. We use geodesic interpolation as a representation validity test in Section 5.2.

**Assumption 2.4.** We adopt standard causal identification assumptions, augmented with mild regularity for Wasserstein risks. (i) *Consistency.* If $T = t$, then $Y = Y(t)$. (ii) *Unconfoundedness.* $\{Y(t)\}_{t \in \mathcal{T}} \perp T \mid X$. (iii) *Overlap / positivity.* For all $x \in \mathcal{X}$ and $t \in \mathcal{T}$, $0 < \mathbb{P}(T = t \mid X = x) < 1$. (iv) *Finite second moments.* $\mathbb{E}\big[\|Y(t)\|^2\big] < \infty$ for all $t$ (ensuring $\mathcal{W}_2$ is well-defined).

# 3. Theoretical Analysis

This section develops the theoretical guarantees underlying the adaptive balancing weight $\widehat{\alpha}(\mathcal{S})$ defined in (7). Our analysis follows the same blueprint as the binary-treatment literature (Johansson et al., 2016; Shalit et al., 2017): (i) establish a generalization bound whose leading terms decompose into a factual component and a representation imbalance component, (ii) turn the bound into an explicit *controlled compression* objective indexed by $\alpha$, and (iii) analyze the statistical behavior of the resulting estimator. The main focus here is the *finite-sample error bound* and *asymptotic normality* of $\widehat{\alpha}(\mathcal{S})$, together with the strategy-dependent scaling in the number of treatments $K$. All proofs are deferred to Appendix C.

## 3.1. A Multi-Treatment Generalization Bound: Decomposition and Strategy Dependence

We first state a generalization bound that extends the binary decomposition in Shalit et al. (2017) to the multi-treatment setting. Recall $\epsilon_{\mathrm{ITE}}(\Phi, h)$ in (2), factual risk $\epsilon_F(\Phi, h)$ in (3), and the strategy-dependent imbalance functional $\mathcal{R}_{\mathcal{S}}(\Phi)$ introduced in Section 2.

**Assumption 3.1** (Regularity for counterfactual generalization). Assume the following hold. (i) (*Lipschitz prediction.*) For each $t \in \mathcal{T}$, the conditional predictor $z \mapsto h(z, t)$ is $L_h$-Lipschitz (under a metric compatible with $\mathcal{W}_2$). (ii) (*Bounded loss.*) The prediction loss $\ell(\widehat{P}, y)$ is bounded by $M$ and is $L_\ell$-Lipschitz in $\widehat{P}$. (iii) (*IPM/HSIC function class.*) The discrepancy class $\mathcal{G}$ defining $\mathrm{IPM}_{\mathcal{G}}$ is uniformly bounded, and the kernels used for MMD/HSIC are bounded. (iv) (*Identification assumptions.*) Consistency, unconfoundedness, and overlap hold (Assumption 2.4 in Section 2).

**Lemma 3.2** (Multi-treatment generalization bound (schematic form)). *Under Assumption 3.1, for any fixed strategy $\mathcal{S}$ and any $(\Phi, h)$ in the model class, with*

*probability at least $1 - \delta$, $\epsilon_{\mathrm{ITE}}(\Phi, h) \leq C_F \epsilon_F(\Phi, h) + C_B \mathcal{R}_{\mathcal{S}}(\Phi) + C_C \mathrm{Complexity}(h \circ \Phi; n, \delta)$, where $C_F, C_B, C_C > 0$ depend only on regularity parameters (e.g., $L_h, L_\ell, M$ and the discrepancy function class) and do not depend on $n$. Moreover, $\mathcal{R}_{\mathcal{S}}(\Phi)$ specializes to the binary discrepancy term when $K = 2$ and $\mathcal{S} = \mathrm{pair}$, recovering the structure of Shalit et al. (2017).*

Lemma 3.2 formalizes the central statistical trade-off of causal representation learning: ITE generalization is controlled by (i) predictive fidelity on factual data and (ii) representation-level imbalance across treatments. It *does not* claim that a particular coefficient ratio is known or identifiable in practice. Instead, it motivates selecting an optimal point on the Pareto frontier of factual fit versus invariance. This is precisely the controlled compression viewpoint encoded by (4)–(5) and the bilevel estimator (7).

## 3.2. Finite-Sample Accuracy of $\widehat{\alpha}(\mathcal{S})$

We now analyze the statistical behavior of the bound-optimized weight $\widehat{\alpha}(\mathcal{S})$. For clarity, we make explicit the *profile* criterion induced by (7). Fix a strategy $\mathcal{S}$ and a compact search range $\mathcal{A} = [\alpha_{\min}, \alpha_{\max}]$. Define the empirical profile bound

$$\widehat{Q}_{\mathcal{S}}(\alpha) := \inf_{\theta} \left\{ \widehat{\epsilon}_F(\theta) + \alpha \widehat{\mathcal{R}}_{\mathcal{S}}(\theta) \right\} + \mathrm{Comp}_{\mathcal{S}}(\alpha; n, \delta), \ \alpha \in \mathcal{A}, \tag{8}$$

and its population analogue

$$Q_{\mathcal{S}}(\alpha) := \inf_{\theta} \left\{ \epsilon_F(\theta) + \alpha \mathcal{R}_{\mathcal{S}}(\theta) \right\} + \mathrm{Comp}_{\mathcal{S}}(\alpha; n, \delta). \tag{9}$$

We denote the *bound-optimal* weight by

$$\alpha_{\mathcal{S}}^{\mathrm{bd}}(n) \in \arg\min_{\alpha \in \mathcal{A}} Q_{\mathcal{S}}(\alpha), \qquad \widehat{\alpha}_{\mathcal{S}} \in \arg\min_{\alpha \in \mathcal{A}} \widehat{Q}_{\mathcal{S}}(\alpha), \tag{10}$$

where we emphasize that $\alpha_{\mathcal{S}}^{\mathrm{bd}}(n)$ may be sample-size dependent since $\mathrm{Comp}_{\mathcal{S}}(\alpha; n, \delta)$ typically scales as $O(n^{-1/2})$. *Remark.* The quantity $\alpha_{\mathcal{S}}^{\mathrm{bd}}(n)$ is the principled target of our estimator: it minimizes a valid upper bound on ITE risk. This is conceptually distinct from the unobservable risk-optimal parameter in (6), but yields an oracle-type guarantee through bound minimization (Corollary 3.6 below). To state finite-sample and asymptotic results, we characterize first-order optimality via an envelope argument.

**Lemma 3.3** (Profile score for $\alpha$). *Assume that for each $\alpha \in \mathcal{A}$, the minimizers in (8) and (9) exist and admit measurable selections $\widehat{\theta}(\alpha) \in \arg\min_{\theta}\{\widehat{\epsilon}_F(\theta) + \alpha\widehat{\mathcal{R}}_{\mathcal{S}}(\theta)\}$ and $\theta^{\star}(\alpha) \in \arg\min_{\theta}\{\epsilon_F(\theta) + \alpha\mathcal{R}_{\mathcal{S}}(\theta)\}$. Assume further that $\mathrm{Comp}_{\mathcal{S}}(\alpha; n, \delta)$ is differentiable in $\alpha$. Then $\widehat{Q}_{\mathcal{S}}$ and $Q_{\mathcal{S}}$ are directionally differentiable and satisfy $\widehat{Q}'_{\mathcal{S}}(\alpha) = \widehat{\mathcal{R}}_{\mathcal{S}}(\widehat{\theta}(\alpha)) + \partial_{\alpha}\mathrm{Comp}_{\mathcal{S}}(\alpha; n, \delta)$, $Q'_{\mathcal{S}}(\alpha) = \mathcal{R}_{\mathcal{S}}(\theta^{\star}(\alpha)) + \partial_{\alpha}\mathrm{Comp}_{\mathcal{S}}(\alpha; n, \delta)$. In particular, any interior minimizer $\widehat{\alpha}_{\mathcal{S}} \in (0, \alpha_{\max})$ satisfies $\widehat{Q}'_{\mathcal{S}}(\widehat{\alpha}_{\mathcal{S}}) = 0$.*

Lemma 3.3 reduces estimation of $\alpha$ to controlling the deviation of the empirical imbalance term evaluated at the $\alpha$-dependent optimizer. We state a high-probability bound that makes the strategy dependence explicit.

**Assumption 3.4** (Curvature and uniform concentration). Assume the following. (i) (*Strong convexity in $\alpha$.*) The population profile criterion $Q_{\mathcal{S}}(\alpha)$ is twice differentiable on $\mathcal{A}$ and satisfies $\inf_{\alpha \in \mathcal{A}} Q_{\mathcal{S}}''(\alpha) \geq \kappa_{\mathcal{S}} > 0$. (ii) (*Uniform concentration of imbalance.*) There exists a nonnegative function $r_{\mathcal{S}}(n, \delta, K)$ such that, with probability at least $1 - \delta$,
$$\sup_{\alpha \in \mathcal{A}} \left| \widehat{\mathcal{R}}_{\mathcal{S}}(\widehat{\theta}(\alpha)) - \mathcal{R}_{\mathcal{S}}(\theta^{\star}(\alpha)) \right| \leq r_{\mathcal{S}}(n, \delta, K).$$

**Theorem 3.5** (Finite-sample deviation bound for $\widehat{\alpha}_{\mathcal{S}}$). *Under Assumption 3.4, any minimizer $\widehat{\alpha}_{\mathcal{S}}$ in (10) satisfies, with probability at least $1 - \delta$, $\left| \widehat{\alpha}_{\mathcal{S}} - \alpha_{\mathcal{S}}^{\mathrm{bd}}(n) \right| \leq r_{\mathcal{S}}(n, \delta, K)/\kappa_{\mathcal{S}}$. For different instantiations of $\widehat{\mathcal{R}}_{\mathcal{S}}$, $r_{\mathcal{S}}(n, \delta, K)$ typically admits the scaling $r_{\mathrm{pair}}(n, \delta, K) = \mathcal{O}\left( K^2 \sqrt{\log(1/\delta)}/\sqrt{n} \right)$, $r_{\mathrm{ova}}(n, \delta, K) = \mathcal{O}\left( K \sqrt{\log(1/\delta)}/\sqrt{n} \right)$, $r_{\mathrm{agg}}(n, \delta, K) = \mathcal{O}\left( \sqrt{\log(1/\delta)}/\sqrt{n} \right)$, where constants depend only on kernel bounds and moment conditions instead of $K$.*

Theorem 3.5 gives a direct, interpretable message: *hyperparameter selection becomes a statistical estimation problem.* For instance, in dose-response estimation with $K = 50$ dosages, pairwise balancing aggregates on the order of $K^2 \approx 2500$ discrepancy terms. Even if each term concentrates at the standard $\mathcal{O}(n^{-1/2})$ rate, the aggregation leads to an $\mathcal{O}(K^2/\sqrt{n})$ deviation in the tuning signal and thus an $\mathcal{O}(K^2/\sqrt{n})$ error in $\widehat{\alpha}$. Keeping the tuning error fixed therefore requires $n = \Omega(K^4)$ samples, which is precisely the instability observed in large-$K$ regimes. In contrast, aggregation controls imbalance through a *single* dependence functional (HSIC), yielding an $\mathcal{O}(n^{-1/2})$ tuning error independent of $K$.

A common misunderstanding is to interpret $\widehat{\alpha}$ as approximating a universal constant such as $C_B/C_F$ in Lemma 3.2. Theorem 3.5 does *not* require such constants to be identifiable. Instead, it guarantees that $\widehat{\alpha}$ is close to the (sample-size dependent) minimizer of a *valid* upper bound on ITE risk. This is a different and stronger notion of justification than heuristic tuning, because it directly controls out-of-sample ITE error through a provable bound[1].

**Corollary 3.6** (Oracle bound guarantee). *Let $\widehat{\alpha}_{\mathcal{S}}$ be defined by (10). If $\sup_{\alpha \in \mathcal{A}} |\widehat{Q}_{\mathcal{S}}(\alpha) - Q_{\mathcal{S}}(\alpha)| \leq \eta_n$ with probability*

---

[1]Crucially, validity of the bound does not rely on identifying a specific coefficient such as $C_B/C_F$. The generalization result implies the existence of a *family* of upper bounds indexed by $\alpha$: for all sufficiently large $\alpha$, the inequality holds uniformly over $\theta$. Our procedure searches within this family and selects the tightest bound using empirical surrogates. Therefore, $\widehat{\alpha}$ does not approximate an unknown constant, but rather indexes a valid member of the bound family whose numerical value is optimal for the given sample.

*at least $1 - \delta$, then*
$$Q_{\mathcal{S}}(\widehat{\alpha}_{\mathcal{S}}) \leq \inf_{\alpha \in \mathcal{A}} Q_{\mathcal{S}}(\alpha) + 2\eta_n \text{ with probability at least } 1 - \delta.$$

### 3.3. Asymptotic Normality and Stability

We now establish the asymptotic distribution of $\widehat{\alpha}_{\mathcal{S}}$ via central limit theorem. The proof uses standard M-estimation arguments applied to the profile score in Lemma 3.3.

**Assumption 3.7** (Asymptotic differentiability and CLT for the profile score). Assume: (i) $\alpha_{\mathcal{S}}^{\mathrm{bd}}(n) \to \alpha_{\mathcal{S}}^{\infty} \in (\alpha_{\min}, \alpha_{\max})$ and $Q_{\mathcal{S}}''(\alpha)$ is continuous at $\alpha_{\mathcal{S}}^{\infty}$ with $Q_{\mathcal{S}}''(\alpha_{\mathcal{S}}^{\infty}) > 0$. (ii) The centered profile score admits a CLT: $\sqrt{n}\left( \widehat{Q}_{\mathcal{S}}'(\alpha_{\mathcal{S}}^{\infty}) - Q_{\mathcal{S}}'(\alpha_{\mathcal{S}}^{\infty}) \right) \Rightarrow \mathcal{N}(0, \sigma_{\mathcal{S}}^2)$, for some $\sigma_{\mathcal{S}}^2 \in (0, \infty)$, and $\widehat{Q}_{\mathcal{S}}'$ converges uniformly to $Q_{\mathcal{S}}'$ on a neighborhood of $\alpha_{\mathcal{S}}^{\infty}$.

Assumption 3.7 is standard in the asymptotic theory of profile M-estimation and tuning-parameter selection. Interiority of the minimizer, local curvature, and a central limit theorem for the profile score are classical conditions ensuring asymptotic normality of one-dimensional profile estimators (van der Vaart, 1998; van der Vaart & Wellner, 1996; Kosorok, 2008). In our setting, the profile score reduces to an empirical imbalance functional evaluated at a data-dependent optimizer; for IPM- and HSIC-based constructions this functional is a U- or V-statistic, for which central limit theorems and uniform convergence results are well established (Serfling, 1980; Gretton et al., 2005). These conditions are empirically diagnosable (e.g., via curvature of the empirical profile criterion and variance estimation of the score), robust to mild violations (boundary optima or weak curvature alter the limiting distribution but not consistency), and logically necessary for any asymptotic normal approximation. Thus, Assumption 3.7 does not introduce ad hoc restrictions, but places $\widehat{\alpha}$ squarely within a well-understood class of statistical estimators.

**Theorem 3.8** (Asymptotic normality of $\widehat{\alpha}_{\mathcal{S}}$). *Under Assumption 3.7, $\sqrt{n}(\widehat{\alpha}_{\mathcal{S}} - \alpha_{\mathcal{S}}^{\infty}) \Rightarrow \mathcal{N}\left( 0, \dfrac{\sigma_{\mathcal{S}}^2}{\left( Q_{\mathcal{S}}''(\alpha_{\mathcal{S}}^{\infty}) \right)^2} \right).$ Consequently, $\widehat{\alpha}_{\mathcal{S}}$ admits asymptotically valid Wald-type confidence intervals once $\sigma_{\mathcal{S}}^2$ is consistently estimated.*

**Corollary 3.9** (Stability scaling with $K$). *Under mild dependence conditions across treatment arms, the asymptotic variance as given above scales as $\mathrm{Var}(\widehat{\alpha}_{\mathrm{pair}}) = \Theta(K^4/n)$, $\mathrm{Var}(\widehat{\alpha}_{\mathrm{ova}}) = \Theta(K^2/n)$, $\mathrm{Var}(\widehat{\alpha}_{\mathrm{agg}}) = \Theta(1/n)$. where $\Theta(\cdot)$ hides constants depending on kernel bounds, overlap constants, and regularity of the representation class, but not on $K$.*

Theorem 3.8 clarifies that under our bound-driven formulation, $\widehat{\alpha}$ is a one-dimensional M-estimator defined by an explicit optimization criterion. It therefore inherits the same

**Algorithm 1** Bound-Optimized Adaptive Balancing (BOAB)

**Require:** Data $\mathcal{D} = \{(X_i, T_i, Y_i)\}_{i=1}^n$, strategy $\mathcal{S} \in \{\text{pair}, \text{ova}, \text{agg}\}$, search set $\mathcal{A} \subset [\alpha_{\min}, \alpha_{\max}]$, confidence level $\delta$.

1: **for** $\alpha \in \mathcal{A}$ **do**
2:     Train $\widehat{\theta}(\alpha, \mathcal{S}) \in \arg\min_\theta \widehat{\epsilon}_F(\theta) + \alpha \widehat{\mathcal{R}}_\mathcal{S}(\theta)$.
3:     Compute bound value $\widehat{Q}_\mathcal{S}(\alpha) = \widehat{\epsilon}_F(\widehat{\theta}(\alpha, \mathcal{S})) + \alpha \widehat{\mathcal{R}}_\mathcal{S}(\widehat{\theta}(\alpha, \mathcal{S})) + \text{Comp}_\mathcal{S}(\alpha; n, \delta)$.
4: **end for**
5: Output $\widehat{\alpha}_\mathcal{S} \in \arg\min_{\alpha \in \mathcal{A}} \widehat{Q}_\mathcal{S}(\alpha)$ and $\widehat{\theta}(\widehat{\alpha}_\mathcal{S}, \mathcal{S})$.

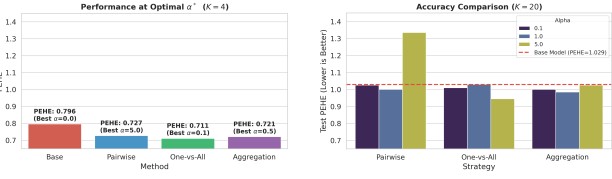

*(a)* Medium-Scale ($K = 4$)     *(b)* Large-Scale ($K = 20$)

*Figure 1.* **Performance Comparison (PEHE, lower is better).** Both panels share a unified $y$-axis for direct comparison. (a) At $K = 4$, all balancing strategies outperform the unadjusted Base Model. (b) At $K = 20$, Pairwise degrades catastrophically under strong regularization while Aggregation remains robust; the Base Model (PEHE: 1.029) is included as reference. (See Appendix D.3, Figure 5 for training efficiency; Appendix E.1, Table 1 for comparison with KNN/IPW baselines.)

statistical structure as classical estimators: a finite-sample deviation bound (Theorem 3.5) and a CLT (Theorem 3.8). The contribution is not that $\alpha$ exists, but that its instability can be quantified and traced back to the *structure of imbalance estimation*.

We summarize the resulting procedure in Algorithm 1. The algorithm is agnostic to the representation architecture (discriminative or generative) and only requires the ability to (i) train $\theta$ for a given $\alpha$ and strategy $\mathcal{S}$, and (ii) evaluate $\widehat{\mathcal{R}}_\mathcal{S}$ and $\text{Comp}_\mathcal{S}(\alpha; n, \delta)$. Algorithm 1 can be instantiated either by a coarse-to-fine grid over $\alpha$ (robust and simple) or by one-dimensional derivative-free search. Theoretical results in Theorems 3.5–3.8 apply to either implementation as long as $\widehat{\alpha}_\mathcal{S}$ is an approximate minimizer of $\widehat{Q}_\mathcal{S}$ with optimization error negligible compared to statistical error.

## 4. Experiments

We conduct a systematic empirical evaluation to validate our proposed multi-treatment balancing framework. Specifically, we address three research questions: **(1) Necessity:** Is representation compression strictly necessary over classical matching and weighting baselines? **(2) Mechanism:** How does $\alpha$ mediate the bias-variance trade-off? **(3) Scalability & Robustness:** Does the Aggregation strategy remain stable and accurate across large-scale and high-dimensional settings?

**Experimental Setup.** To evaluate robustness under severe selection bias, we generate semi-synthetic datasets characterized by high-dimensional covariates and complex treatment interactions. We benchmark four methods: (1) **Base Model**: a standard **Counterfactual Regression** (CFR) architecture trained solely on the factual prediction loss (setting $\alpha = 0$), serving as the unadjusted baseline; (2) **Pairwise (CFR-Pair)**: representation learning with $\mathcal{O}(K^2)$ pairwise MMD constraints across all treatment pairs (Schwab et al., 2018); (3) **One-vs-All (CFR-OVA)**: representation learning with $\mathcal{O}(K)$ one-vs-all MMD constraints; (4) **Treatment Aggregation (CFR-Agg, Ours)**: our proposed $\mathcal{O}(1)$ HSIC-based

global independence constraint with learnable treatment embeddings. All three CFR variants use the bound-optimized $\widehat{\alpha}$ from Algorithm 1. The primary metric is PEHE (lower is better). Full dataset generation details are provided in Appendix D.1.

We further conduct four robustness stress tests in Appendix E: (i) a "Double Curse" experiment ($N = 1000$, $D = 2000$, $K = 10$) with up to 99.5% noise dimensions (Appendix E.2); (ii) a proxy/noisy confounder test under increasing measurement noise (Appendix E.3); (iii) a Mechanism Heterogeneity Stress Test controlling structural mismatch across treatments (Appendix E.4); (iv) an extreme-scale test at $K = 100$ (Appendix E.5). We additionally compare against classical KNN Matching and IPW baselines in Appendix E.1 to confirm the necessity of representation compression.

### 4.1. Evaluation on Medium-Scale Scenarios

**Efficacy of Balancing Strategies.** As shown in Figure 1(a), all proposed balancing strategies outperform the unadjusted Base Model (PEHE 0.796) in the $K = 4$ regime. The **One-vs-All (OVA)** strategy achieves the best performance with the lowest error (PEHE 0.711). This confirms that for relatively small action spaces, decomposing the multi-treatment problem into multiple "Target vs. Rest" tasks provides a highly robust optimization landscape for representation learning. The **Pairwise** and **Aggregation (Agg-T)** strategies yield comparable results, with PEHE values of 0.727 and 0.721, respectively. While Agg-T shows a slight edge over Pairwise in this medium-scale setting, its primary advantage—decoupling computational complexity from treatment cardinality—becomes more pronounced as $K$ increases (see Section 4.2). The fact that Agg-T matches the performance of more complex constraints even at $K = 4$ demonstrates its viability as a general-purpose balancing strategy.

**Mechanism of $\alpha$ (Bias-Variance Trade-off).** The opti-

mal performance for each method is attained at different levels of the balancing weight $\alpha$, reflecting their inherent sensitivity to the deconfounding-prognostic trade-off: (1) **Optimal Weight Selection:** The OVA strategy achieves its peak performance at a very small weight ($\alpha = 0.1$), indicating that its objective is highly efficient at removing bias with minimal regularization. In contrast, the Pairwise strategy requires a much stronger weight ($\alpha = 5.0$) to achieve its best alignment, suggesting a more difficult optimization path when satisfying $\mathcal{O}(K^2)$ constraints. (2) **Controlled Compression:** The Aggregation strategy finds its balance at a moderate weight ($\alpha = 0.5$). These results empirically validate our theoretical framework: $\alpha$ is not a universal constant but a strategy-dependent parameter that must be optimized to prevent representation collapse while ensuring sufficient distributional invariance. We further validate that our 1D Grid Search reliably finds the true global optimum $\alpha^*$: an empirical comparison against continuous Gradient Descent (Appendix D.4) confirms that both methods converge to the same $\alpha^* = 0.50$ (PEHE: 0.722), while Grid Search additionally offers non-convex robustness and greater computational efficiency.

### 4.2. Scalability Analysis on Large-Scale Scenarios

We further extend the simulation to $K = 20$ to evaluate scalability. Figure 1(b) demonstrates a fundamental divergence in model behavior: As detailed in **Appendix D.3**, the Pairwise strategy exhibits quadratic computational complexity ($\mathcal{O}(K^2)$), resulting in training instability and excessive runtime. In contrast, our **Aggregation strategy** maintains constant-time complexity ($\mathcal{O}(1)$) via the HSIC constraint, achieving stable convergence and competitive accuracy equivalent to small-scale regimes. This empirically validates that our method successfully decouples computational cost from treatment cardinality.

It also reveals the robustness of different strategies. The Pairwise strategy suffers severe degradation under strong regularization ($\alpha = 5.0$), with PEHE spiking above 1.3. We attribute this to the **"Over-constraint"** phenomenon: satisfying 190 conflicting alignment goals restricts the representation space excessively. While **One-vs-All** achieves the lowest absolute error at $\alpha = 5.0$ (PEHE $\approx 0.95$), it incurs a higher computational cost. **Agg-T** offers the best **efficiency-stability trade-off**: it maintains competitive accuracy (PEHE $\approx 1.0$) across all $\alpha$ settings without the volatility of Pairwise methods, proving to be the most scalable paradigm for high-dimensional inference.

## 5. Generative Extension and Geometry

While the discriminative strategies evaluated in Section 4 demonstrate the efficacy of optimal compression, they primarily focus on estimating scalar outcomes. We now ex-

tend our framework to a **generative paradigm**, proposing **Multi-Treatment CausalEGM**. This extension lifts our multi-treatment compression framework from scalar-outcome estimation to *structured counterfactual generation*, and serves two primary purposes: (1) to enable high-dimensional counterfactual generation; and (2) to verify the geometric integrity of the learned representation through geodesic interpolation.

### 5.1. Multi-Treatment CausalEGM

**Architecture.** We build upon CausalEGM (Liu et al., 2024), a bidirectional generative model that disentangles confounding factors $Z_c$ from instrumental factors $Z_t$ (where $Z_c$ and $Z_t$ are CausalEGM's confounding and instrumental latent factors, distinct from the treatment $T$). To address the scalability challenges of high-cardinality treatments, we introduce two structural modifications to the original binary design (the complete architecture is provided in Appendix D.2, Figure 4): (1) **Vectorized Treatment Embeddings:** Instead of one-hot encoding, we map the discrete treatment index $t \in \{0, \dots, K-1\}$ to a dense vector $e_t \in \mathbb{R}^{d_e}$ via a learnable lookup table, enabling the model to capture topological relationships between treatments within the latent space. (2) **Softmax Intervention Mechanism:** We replace the binary generation mechanism with a multi-class Softmax head, $P(T|Z_c) = \text{Softmax}(f_\theta(Z_c))$, allowing the model to approximate complex propensity surfaces over $K$ categories.

**Experimental Setup.** We evaluate the model on the semi-synthetic UCI Digits dataset to test its ability to interpolate causal effects across a structured treatment manifold. (See Appendix D.1 for data generation details.)

**Analysis.** We evaluate the Multi-Treatment CausalEGM against the discriminative baselines established in Section 4. As illustrated in Figure 2(a), the generative model faithfully recovers the non-linear **Average Dose-Response Function (ADRF)**, accurately locating the global minimum at $T = 4$. Quantitatively, it achieves a PEHE of **0.65** (Figure 2(b)), which significantly outperforms the unadjusted Base Model (0.79)—a $\sim 17.7\%$ relative reduction in causal prediction error—and is competitive with the discriminative Aggregation baseline (0.67). It is important to note that generative models such as CausalEGM inherently optimize for holistic distribution matching (via evidence lower bounds or adversarial losses) rather than point-wise MSE; the fact that our simple HSIC aggregation constraint atop this complex generative backbone reduces point-wise PEHE by nearly 18% provides strong empirical evidence of its regularizing power. While the specialized One-vs-All strategy achieves lower scalar error (0.24)—expected due to its purely discriminative focus—our approach maintains robust causal identification while fulfilling the more complex objective of high-dimensional counterfactual generation. Furthermore,

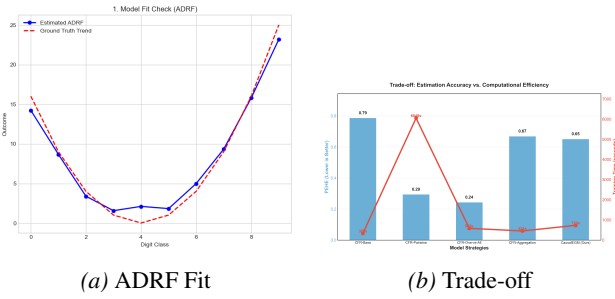

*(a)* ADRF Fit   *(b)* Trade-off

*Figure 2.* **Performance and Efficiency Analysis on Digits Dataset.** (a) The estimated ADRF (blue) closely tracks the ground truth (red). (b) Dual-axis comparison showing PEHE error (bars) and Training Time (line).

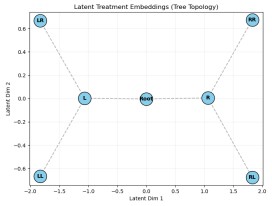
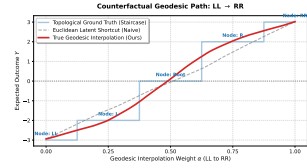

*(a)* Latent Space Topology   *(b)* Geodesic Interpolation

*Figure 3.* **Geometric Validation on Hierarchical Treatments.** (a) The learned embeddings spontaneously recover the underlying tree structure, placing the Root centrally and separating the L/R branches. (b) Corrected counterfactual interpolation from Leaf LL ($Y = -3$) to Leaf RR ($Y = +3$): the **True Geodesic path** (red) precisely passes through $Y = 0$ at $\alpha = 0.5$ (Root node), hits $Y = -2$ at $\alpha = 0.25$ (Node L), and $Y = +2$ at $\alpha = 0.75$ (Node R)—flawlessly recovering all intermediate topological states. The Euclidean shortcut (grey dashed) completely misses this causal structure.

despite the added generative overhead, our model maintains comparable training efficiency to the Aggregation strategy (detailed runtime analysis provided in **Appendix D.3**), confirming that the embedding-based architecture remains scalable for high-dimensional counterfactual generation.

### 5.2. Geometric Fidelity and Topological Consistency

While standard representation learning focuses on predictive accuracy, GCI requires the latent space to recover the underlying geometry of the treatment mechanism. To verify this, we tasked Multi-Treatment CausalEGM with learning a hierarchical treatment structure (a depth-3 binary tree), where causal effects are determined by the node's position in the hierarchy ($Y_{LL} \approx -3, Y_{Root} \approx 0, Y_{RR} \approx +3$).

**Analysis.** Figure 3(a) visualizes the learned latent space. Without access to graph coordinates, our Geodesic-Regularized objective successfully reconstructs the isometric topology: the embedding places the Root (Node 0) centrally, with branches (L/R) and leaves (LL/RR) extending radially. This confirms that the model disentangles struc-

tural relationships from discrete treatment IDs. Furthermore, Figure 3(b) demonstrates the physical plausibility of counterfactual interpolation. When interpolating between distinct sub-types (Leaf LL to Leaf RR), a naive Euclidean baseline (grey dashed line) assumes a linear transition that ignores causal structure. In contrast, our model (red solid line) generates a non-linear path that passes through $Y \approx 0$ at the midpoint ($\alpha = 0.5$). This indicates that the latent interpolation path transitively activates the representation of the common ancestor (Root), mirroring the true causal mechanism rather than taking an impossible "shortcut" through the void. (More details are provided in Appendix D.5, E.6)

## 6. Conclusion

We present a scalable framework for multi-treatment causal representation learning by reframing the balancing objective as *optimal compression*. Our theoretical analysis transforms the balancing weight $\alpha$ from a heuristic hyperparameter into a statistically estimable quantity. To address combinatorial instability, we introduce the One-vs-All and Treatment Aggregation strategies; notably, the latter achieves $\mathcal{O}(1)$ complexity via HSIC, ensuring robustness in high-cardinality treatment regimes. Furthermore, our generative extension (Multi-Treatment CausalEGM) recovers the underlying Wasserstein geodesic geometry of the treatment manifold. Future work will focus on extending these geometric compression principles to continuous treatment spaces and addressing scenarios with latent confounding.

## Acknowledgment

Zhiheng Zhang is supported by "the Fundamental Research Funds for the Central Universities" (Grant No. 2025110602) of Shanghai University of Finance and Economics, and Independent Research Project (Grant No. 2026110081) funded by the School of Statistics and Data Science. This work was supported by the Shanghai Engineering Research Center of Finance Intelligence (Grant No. 19DZ2254600).

## Impact Statement

This paper presents work whose goal is to advance the field of machine learning. There are many potential societal consequences of our work, none of which we feel must be specifically highlighted here.

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

# A. Related Work

**Causal Representation Learning.** The paradigm of learning balanced representations for causal inference was pioneered by Johansson et al. (2016) and Shalit et al. (2017). The central premise is to map covariates into a latent space where treatment groups are distributionally similar (minimizing an IPM), thereby reducing selection bias while retaining prognostic information for outcome prediction. This framework has been extended to various settings, including domain adaptation and instrumental variable scenarios. However, the statistical implications of this "balancing-preservation" trade-off remain a subject of active debate. From an information-theoretic perspective, this trade-off mirrors the Information Bottleneck principle applied to deep learning, where Achille & Soatto (2018) proposed learning optimal representations by controlling the flow of information to minimize sufficiency for the task while maximizing invariance to nuisance factors. While earlier works implicitly assumed that stricter balancing yields better estimators, recent theoretical analyses have highlighted a "no free lunch" phenomenon: aggressive compression to enforce invariance can inadvertently discard outcome-relevant information, potentially increasing the estimation error (Melnychuk et al., 2023). Furthermore, Curth & van der Schaar (2021) scrutinized the architectural inductive biases in this domain, analyzing how the choice between shared representations and separate heads affects generalization across heterogeneous treatment effects. Our work formalizes this trade-off in the multi-treatment regime, transforming the balancing weight from a heuristic hyperparameter into an estimable quantity derived from generalization bounds.

**Distinction from Identifiability-Based Causal Representation Learning.** We use the term "causal representation learning" in the sense of the treatment-effect estimation literature above. It is important to distinguish this from the identifiability-focused field pioneered by Schölkopf et al. (2021), which asks how latent *causal variables* can be recovered from complex observational data—a problem concerned with disentangling independent causal mechanisms at the generative level. Our paper offers a different, more task-focused view: we ask how to learn a *treatment-aware* low-dimensional representation that preserves effect-relevant structure across many related interventions, together with new theory for when this compression remains valid for causal effect estimation and a practical method that scales to complex multi-treatment settings. While both lines of work learn representations for causal purposes, the objectives, assumptions, and technical machinery are distinct; we cite Schölkopf et al. (2021) to acknowledge this important parallel field and to avoid conflating the two meanings of the term.

**Multi-Treatment and Continuous Causal Inference.** Extending causal inference beyond binary treatments introduces significant computational and statistical challenges. Traditional approaches like Generalized Propensity Score (GPS) methods often struggle with high-dimensional covariates. In the representation learning domain, Schwab et al. (2018) and Lopez & Gutman (2017) proposed extensions such as pairwise balancing or multiple-head architectures. Parallel research has also addressed continuous treatment regimes using deep learning; for instance, SCIGAN (Bica et al., 2020) utilizes generative adversarial networks to estimate effects of continuous interventions, while VCNet (Nie et al., 2021) employs varying coefficient neural networks to model continuous dose-response curves. Nevertheless, these discriminative strategies typically incur a computational cost of $\mathcal{O}(K^2)$ to align all pairs of treatment groups, which becomes prohibitive as the number of treatments $K$ increases. Furthermore, treating treatments as disjoint categories ignores the underlying structure (e.g., dosage ordinality) often present in real-world applications. Our proposed *Treatment Aggregation* strategy addresses this scalability bottleneck by reducing the complexity to $\mathcal{O}(1)$ via global independence constraints (HSIC), making representation learning feasible for large-scale action spaces.

**Generative and Geometric Causal Models.** Recent advancements have begun to explore the geometric structure of causal mechanisms. Liu et al. (2024) introduced CausalEGM, a bidirectional generative model that explicitly disentangles confounding and instrumental factors using VAE-GAN architectures. On the theoretical front, Kurisu et al. (2024) proposed Geodesic Causal Inference (GCI), framing causal effect estimation as transport problems on Wasserstein manifolds. Our *Multi-Treatment CausalEGM* bridges these two lines of work. By embedding discrete treatments into a continuous manifold and enforcing geodesic consistency during interpolation, we provide a generative framework that is not only scalable but also physically interpretable, validating that the learned latent space captures the true geometry of the treatment mechanism.

Moreover, this work is also motivated by our previous work (Zhang et al., 2020; Zhang & Su, 2024; Zhang, 2022; Zhang et al., 2023; Zhang, 2024; Su & Zhang et al, 2025; Wang et al., 2025; Zhang et al., 2025; Zhang & Wang, 2025; Li et al., 2026; Su et al., 2023; 2024; 2026; Lu et al., 2026; Zhang, 2026).

# B. Theoretical Supplement

## B.1. Formal Definitions of Discrepancy Measures

We instantiate the discrepancy operators using kernel methods. Let $\mathcal{H}$ be an RKHS with kernel $k(\cdot, \cdot)$. For $\mathcal{S} \in \{pair, ova\}$, we employ the **Maximum Mean Discrepancy (MMD)**, defined as the RKHS distance between mean embeddings: $\text{MMD}(P, Q) := \|\mu_P - \mu_Q\|_{\mathcal{H}}$ (Gretton et al., 2012). This serves as a tractable IPM instantiation where the function class $\mathcal{G}$ is the unit ball in $\mathcal{H}$. For $\mathcal{S} = agg$, we employ the **Hilbert-Schmidt Independence Criterion (HSIC)** to enforce independence. HSIC measures the Hilbert-Schmidt norm of the cross-covariance operator, $\text{HSIC}(Z, E) := \|\mathcal{C}_{ZE}\|_{HS}^2$ (Gretton et al., 2005), which is zero if and only if the representation $\Phi(X)$ and treatment embedding $e(T)$ are statistically independent. We employ standard U-statistic and V-statistic estimators for MMD and HSIC, respectively.

## B.2. Explicit Form of the Complexity Term

To complete the description of the profile bound in (7) of the main text, we provide the explicit formal definition of the complexity term $\text{Comp}_{\mathcal{S}}(\alpha; n, \delta)$. As discussed in Section 2, the penalized objective is equivalent to a constrained optimization problem via Lagrangian duality. We leverage this duality to define the $\alpha$-induced hypothesis class.

**The $\alpha$-Induced Hypothesis Class.** Let $\rho : \mathbb{R}_{\geq 0} \to \mathbb{R}_{\geq 0}$ be the mapping from the Lagrange multiplier $\alpha$ to the effective imbalance budget. From convex duality principles, $\rho(\alpha)$ is a monotonically decreasing function of $\alpha$: a larger penalty weight enforces a stricter constraint on the representation imbalance.

We define the $\alpha$-*induced hypothesis class* $\mathcal{H}_\alpha$ as the set of all predictor-representation pairs that satisfy the imbalance budget implied by $\alpha$:

$$\mathcal{H}_\alpha := \left\{ f = h \circ \Phi \mid \Phi \in \mathbf{\Phi}, h \in \mathbf{H}, \hat{\mathcal{R}}_{\mathcal{S}}(\Phi) \leq \rho(\alpha) \right\}, \tag{11}$$

where $\mathbf{\Phi}$ and $\mathbf{H}$ are the unrestricted function spaces for the representation and predictor, respectively. Crucially, because $\rho(\alpha)$ decreases as $\alpha$ increases, we have the following nested property:

$$\alpha_1 < \alpha_2 \implies \rho(\alpha_1) > \rho(\alpha_2) \implies \mathcal{H}_{\alpha_2} \subseteq \mathcal{H}_{\alpha_1}. \tag{12}$$

This containment relationship directly implies that the complexity of the hypothesis class decreases as the penalty weight $\alpha$ increases.

**Explicit Definition of $\text{Comp}_{\mathcal{S}}$.** We instantiate the complexity term using *Rademacher Complexity*, which measures the richness of the hypothesis class. The term $\text{Comp}_{\mathcal{S}}(\alpha; n, \delta)$ in Eq. (7) is explicitly defined as:

$$\text{Comp}_{\mathcal{S}}(\alpha; n, \delta) := 2\mathfrak{R}_n(\ell \circ \mathcal{H}_\alpha) + M\sqrt{\frac{\log(1/\delta)}{2n}}, \tag{13}$$

where:

- $M$ is the uniform upper bound of the loss function $\ell$.

- $\mathfrak{R}_n(\ell \circ \mathcal{H}_\alpha)$ is the empirical Rademacher complexity of the loss function class composed with the $\alpha$-constrained hypothesis class:

$$\mathfrak{R}_n(\ell \circ \mathcal{H}_\alpha) = \mathbb{E}_\sigma \left[ \sup_{f \in \mathcal{H}_\alpha} \frac{1}{n} \sum_{i=1}^{n} \sigma_i \ell(f(X_i, T_i), Y_i) \right], \tag{14}$$

  with $\sigma_i$ being i.i.d. Rademacher variables taking values $\{-1, +1\}$.

Since $\mathcal{H}_\alpha$ shrinks as $\alpha$ increases (i.e., $\mathcal{H}_{\alpha_2} \subseteq \mathcal{H}_{\alpha_1}$ for $\alpha_2 > \alpha_1$), the supremum in the Rademacher complexity is taken over a strictly smaller set. Therefore, the complexity term satisfies the monotonicity condition required for the trade-off analysis:

$$\frac{\partial}{\partial \alpha} \text{Comp}_{\mathcal{S}}(\alpha; n, \delta) \leq 0. \tag{15}$$

This confirms that stronger compression (larger $\alpha$) reduces the effective capacity of the model class, thereby lowering the generalization gap component of the bound.

**Concrete Examples of $\text{Comp}_{\mathcal{S}}$ Scaling with $\alpha$.** To further illustrate the inverse relationship between $\alpha$ and the complexity term, we provide two concrete examples where the geometry of $\mathcal{H}_\alpha$ can be explicitly characterized.

**Example 1: Effective Dimension in Linear Representation Learning.** Consider a linear representation map $\Phi(x) = Wx$ where $W \in \mathbb{R}^{d \times d_x}$. Suppose the imbalance metric $\hat{\mathcal{R}}_S(\Phi)$ penalizes the projection of data onto treatment-discriminative directions (e.g., maximizing independence). A strict imbalance constraint $\hat{\mathcal{R}}_S(\Phi) \leq \rho(\alpha)$ forces the weight matrix $W$ to be approximately orthogonal to the subspace spanned by the treatment mechanisms.

We can model the effective hypothesis class $\mathcal{H}_\alpha$ as a ball in a subspace of reduced dimension $d_{\text{eff}}(\alpha)$. As $\alpha \to \infty$ (strong compression), $W$ is forced to project data onto the null space of the treatment correlations, reducing the rank of $W$. The Rademacher complexity for such linear classes is bounded by:

$$\mathfrak{R}_n(\ell \circ \mathcal{H}_\alpha) \leq \frac{C}{\sqrt{n}} \sqrt{d_{\text{eff}}(\alpha)}, \tag{16}$$

where $d_{\text{eff}}(\alpha)$ is the effective dimension (e.g., squared Frobenius norm or rank) allowed under the budget $\rho(\alpha)$. Since $\rho(\alpha)$ is decreasing, $d_{\text{eff}}(\alpha)$ decreases as $\alpha$ increases, explicitly reducing the complexity term.

**Example 2: Covering Numbers and Lipschitz Constraints.** Consider the case where the representation $\Phi$ is parameterized by a neural network, and the hypothesis class $\mathcal{H}_\alpha$ is characterized by its Lipschitz constant $L_\Phi$. Enforcing distributional invariance (small $\hat{\mathcal{R}}_S$) requires the representation to map distinct treatment groups to overlapping regions in the latent space. This "squeezing" effect often necessitates a smaller Lipschitz constant to suppress the variation due to treatment-correlated covariates.

Let $L(\alpha)$ be the maximum Lipschitz constant permissible under the constraint $\hat{\mathcal{R}}_S(\Phi) \leq \rho(\alpha)$. As $\alpha$ increases, the allowable class of functions becomes smoother, implying $L(\alpha_2) \leq L(\alpha_1)$ for $\alpha_2 > \alpha_1$. Using the Dudley's entropy integral bound for Rademacher complexity:

$$\mathfrak{R}_n(\ell \circ \mathcal{H}_\alpha) \leq \inf_{\epsilon > 0} \left( 4\epsilon + \frac{12}{\sqrt{n}} \int_\epsilon^M \sqrt{\log \mathcal{N}(\tau, \mathcal{H}_\alpha, \|\cdot\|_\infty)} d\tau \right). \tag{17}$$

Since the covering number $\mathcal{N}(\tau, \mathcal{H}_\alpha, \cdot)$ scales polynomially with the Lipschitz constant $L(\alpha)$ (i.e., $\log \mathcal{N} \propto L(\alpha)^d$), a smaller $L(\alpha)$ resulting from stronger compression directly yields a tighter complexity bound.

# C. Proofs for Section 3

## C.1. Preliminaries and auxiliary tools

**Basic notation.** Let $\mathcal{T} = \{0, \ldots, K-1\}$, and write $\pi_t := \mathbb{P}(T = t)$, $\pi_{\min} := \min_{t \in \mathcal{T}} \pi_t$. For a representation $\Phi : \mathcal{X} \to \mathcal{Z}$, denote by $P_\Phi^{(t)} := \mathcal{L}(\Phi(X) \mid T = t)$ the conditional distribution of $\Phi(X)$ given $T = t$. For a measurable map $h : \mathcal{Z} \times \mathcal{T} \to \mathcal{P}_2(\mathcal{Y})$, let $\widehat{P}_{Y|z,t} = h(z, t)$.

**IPM, MMD, HSIC.** Given a function class $\mathcal{G}$ on $\mathcal{Z}$, define the integral probability metric

$$\text{IPM}_\mathcal{G}(P, Q) := \sup_{g \in \mathcal{G}} \left| \mathbb{E}_P[g] - \mathbb{E}_Q[g] \right|.$$

A standard instantiation is MMD: let $\mathcal{H}$ be an RKHS on $\mathcal{Z}$ with reproducing kernel $k$ and unit ball $\mathcal{G} = \{g \in \mathcal{H} : \|g\|_\mathcal{H} \leq 1\}$. Then

$$\text{MMD}_\mathcal{H}(P, Q) = \text{IPM}_\mathcal{G}(P, Q) = \|\mu_P - \mu_Q\|_\mathcal{H},$$

where $\mu_P := \mathbb{E}_{Z \sim P}[k(Z, \cdot)]$ is the mean embedding. For treatment aggregation, let $e : \mathcal{T} \to \mathbb{R}^{d_e}$ and $E_T := e(T)$. Given bounded kernels $k$ on $\mathcal{Z}$ and $\ell$ on $\mathbb{R}^{d_e}$, define HSIC by

$$\text{HSIC}(Z, E) := \|C_{ZE}\|_{\text{HS}}^2,$$

the squared Hilbert–Schmidt norm of the cross-covariance operator; $\text{HSIC}(Z, E) = 0$ iff $Z \perp E$ under mild conditions.

**U- and V-statistic estimators.** Assume $k$ is bounded: $\sup_z k(z,z) \leq \kappa^2$ and similarly $\sup_e \ell(e,e) \leq \lambda^2$. For samples $\{Z_i\}_{i=1}^m \sim P$ and $\{W_j\}_{j=1}^n \sim Q$, an unbiased U-statistic estimator of $\mathrm{MMD}^2(P,Q)$ is

$$\widehat{\mathrm{MMD}}^2(P,Q) = \frac{1}{m(m-1)}\sum_{i \neq i'} k(Z_i, Z_{i'}) + \frac{1}{n(n-1)}\sum_{j \neq j'} k(W_j, W_{j'}) - \frac{2}{mn}\sum_{i,j} k(Z_i, W_j).$$

The empirical HSIC is a (degenerate) V-statistic; we denote by $\widehat{\mathrm{HSIC}}_n$ any of the standard estimators (centered Gram-matrix form).

**Uniform generalization via Rademacher complexity.** Let $\mathcal{F}$ be a class of measurable functions bounded in $[-M, M]$. Then with probability at least $1 - \delta$,

$$\sup_{f \in \mathcal{F}} \left| \mathbb{E}[f] - \frac{1}{n}\sum_{i=1}^n f(X_i) \right| \leq 2\Re_n(\mathcal{F}) + M\sqrt{\frac{\log(2/\delta)}{2n}},$$

where $\Re_n(\mathcal{F})$ is the empirical (or expected) Rademacher complexity. We will use this as a black box to justify the generic complexity term in Lemma 3.2.

### C.2. Proof of Lemma 3.2

Lemma 3.2 is stated in a schematic form to accommodate (i) scalar-outcome PEHE and (ii) distributional estimands in Wasserstein space. We give a self-contained proof template that makes the dependence on a representation discrepancy explicit and reduces to the binary argument of Shalit et al. (2017) when $K = 2$.

**Step 1: Reducing distributional ITE error to per-treatment outcome estimation error.** Fix $x \in \mathcal{X}$ and two treatments $j, k$. Let $P_{j,x} := P_{Y(j)|X=x}$ and $\widehat{P}_{j,x} := \widehat{P}_{Y|\Phi(x),j}$ (and similarly for $k$). By the reverse triangle inequality of $\mathcal{W}_2$,

$$\left| \mathcal{W}_2(\widehat{P}_{j,x}, \widehat{P}_{k,x}) - \mathcal{W}_2(P_{j,x}, P_{k,x}) \right| \leq \mathcal{W}_2(\widehat{P}_{j,x}, P_{j,x}) + \mathcal{W}_2(\widehat{P}_{k,x}, P_{k,x}).$$

Squaring and using $(a+b)^2 \leq 2a^2 + 2b^2$ yields

$$\left( \widehat{\tau}_{j,k}(x) - \tau_{j,k}(x) \right)^2 \leq 2\,\mathcal{W}_2(\widehat{P}_{j,x}, P_{j,x})^2 + 2\,\mathcal{W}_2(\widehat{P}_{k,x}, P_{k,x})^2. \tag{18}$$

Summing (18) over all $0 \leq j < k \leq K - 1$ and taking expectation over $X$ gives

$$\epsilon_{\mathrm{ITE}}(\Phi, h) \leq 2(K-1) \sum_{t=0}^{K-1} \mathbb{E}_X \left[ \mathcal{W}_2(\widehat{P}_{t,X}, P_{t,X})^2 \right]. \tag{19}$$

Thus, controlling ITE risk reduces to controlling *per-treatment potential-outcome distribution estimation errors*.

**Step 2: From potential-outcome error to (factual) prediction plus domain discrepancy.** For each $t \in \mathcal{T}$, define the "target" risk for predicting $Y(t)$ under the marginal covariate distribution:

$$\epsilon_{\mathrm{tar}}^{(t)}(\Phi, h) := \mathbb{E}_X \left[ \mathcal{W}_2(\widehat{P}_{t,X}, P_{t,X})^2 \right].$$

Under unconfoundedness and overlap, $\epsilon_{\mathrm{tar}}^{(t)}$ is identified but is not directly estimable without reweighting. The representation-learning strategy controls generalization across treatment arms by bounding differences of risks across *domains* $P_\Phi^{(t)}$ using an IPM (domain-adaptation step).

To keep the argument aligned with the existing binary literature, consider a generic bounded loss $\tilde{\ell}_t(z) := \tilde{\ell}(h(z, t), \cdot)$ whose expectation equals $\epsilon_{\mathrm{tar}}^{(t)}$ up to constants; Assumption 3.1(i)–(ii) implies the map $z \mapsto \mathbb{E}[\tilde{\ell}_t(z) \mid \Phi(X) = z]$ is $L$-Lipschitz for some $L$ depending only on $(L_h, L_\ell)$. Then by the definition of $\mathrm{IPM}_\mathcal{G}$ over a class containing these Lipschitz functions (or by a standard "discrepancy distance" argument as in Johansson et al., 2016), for any two treatment arms $j, k$ we obtain

$$\epsilon_{\mathrm{tar}}^{(k)}(\Phi, h) \leq c_1\, \epsilon_{\mathrm{src}}^{(j)}(\Phi, h) + c_2\, \mathrm{IPM}_\mathcal{G}\left( P_\Phi^{(j)}, P_\Phi^{(k)} \right) + c_3, \tag{20}$$

where $\epsilon_{\mathrm{src}}^{(j)}$ is the (factual) risk on domain $T = j$, and $c_1, c_2, c_3$ depend only on regularity constants and boundedness. In the binary case, (20) is precisely the step exploited in Shalit et al. (2017).

**Step 3: Summation over multiple treatments and strategy dependence.** Summing (20) over all relevant pairs (and using the symmetry $j \leftrightarrow k$) yields an upper bound of the form

$$\sum_{t=0}^{K-1} \epsilon_{\text{tar}}^{(t)}(\Phi, h) \leq C_F' \, \epsilon_F(\Phi, h) + C_B' \, \mathcal{D}_{\text{pair}}\big(\{P_\Phi^{(t)}\}_{t \in \mathcal{T}}\big) + C_0,$$

where $\mathcal{D}_{\text{pair}}(\cdot) = \sum_{j<k} \text{IPM}_{\mathcal{G}}(P_\Phi^{(j)}, P_\Phi^{(k)})$ corresponds to the pairwise strategy. Other strategies correspond to other discrepancy operators $\mathcal{D}_{\mathcal{S}}$:

- For $\mathcal{S} = \text{ova}$, replace each pairwise discrepancy by one-vs-all discrepancies and use triangle/mixture inequalities to obtain a bound with $\mathcal{D}_{\text{ova}}$ (up to constants).

- For $\mathcal{S} = \text{agg}$, $\mathcal{D}_{\text{agg}}$ is a dependence functional (HSIC) satisfying $\mathcal{D}_{\text{agg}} = 0$ iff $\Phi(X) \perp E_T$, hence it enforces global balance; the resulting bound is written with $\mathcal{R}_{\text{agg}}$ as the imbalance term by definition of the strategy-dependent functional in the statement.

Combining with (19) gives

$$\epsilon_{\text{ITE}}(\Phi, h) \leq C_F \, \epsilon_F(\Phi, h) + C_B \, \mathcal{R}_{\mathcal{S}}(\Phi) + C_0, \tag{21}$$

for constants $C_F, C_B, C_0$ depending only on regularity.

**Step 4: Adding the complexity term.** To obtain a high-probability statement for empirical learning, add and subtract $\widehat{\epsilon}_F$ and apply uniform generalization bounds for the composed class $\{(x, t) \mapsto \ell(h(\Phi(x), t), \cdot)\}$ (bounded by $M$). This yields a remainder term of the form $\text{Complexity}(h \circ \Phi; n, \delta)$, e.g., a Rademacher-complexity bound plus a concentration term. Absorbing constants into $C_C$ yields the claimed statement of Lemma 3.2. $\qquad\square$

### C.3. Proof of Lemma 3.3 (profile score)

We prove the score representation using Danskin's theorem / envelope arguments.

Let

$$F(\theta, \alpha) := \widehat{\epsilon}_F(\theta) + \alpha \, \widehat{\mathcal{R}}_{\mathcal{S}}(\theta) \quad \text{and} \quad \widehat{Q}_{\mathcal{S}}(\alpha) = \inf_\theta F(\theta, \alpha) + \text{Comp}_{\mathcal{S}}(\alpha; n, \delta).$$

Fix $\alpha \in \mathcal{A}$ and let $\widehat{\theta}(\alpha) \in \arg\min_\theta F(\theta, \alpha)$ be a measurable selection (assumed to exist). For any direction $u \in \mathbb{R}$, the directional derivative of the infimum satisfies (by Danskin's theorem; see, e.g., Rockafellar & Wets, 1998, Appendix D)

$$\mathrm{d}\widehat{Q}_{\mathcal{S}}(\alpha; u) = \min_{\theta \in \arg\min_\vartheta F(\vartheta, \alpha)} \partial_\alpha F(\theta, \alpha) \, u + \partial_\alpha \text{Comp}_{\mathcal{S}}(\alpha; n, \delta) \, u.$$

Since $\partial_\alpha F(\theta, \alpha) = \widehat{\mathcal{R}}_{\mathcal{S}}(\theta)$ and we have selected $\widehat{\theta}(\alpha)$, we obtain

$$\widehat{Q}_{\mathcal{S}}'(\alpha) = \widehat{\mathcal{R}}_{\mathcal{S}}(\widehat{\theta}(\alpha)) + \partial_\alpha \text{Comp}_{\mathcal{S}}(\alpha; n, \delta),$$

whenever $\widehat{Q}_{\mathcal{S}}$ is differentiable at $\alpha$ (directional differentiability holds under the stated conditions). The population identity is identical with hats removed. Finally, if $\widehat{\alpha}_{\mathcal{S}}$ is an interior minimizer and $\widehat{Q}_{\mathcal{S}}$ is differentiable at $\widehat{\alpha}_{\mathcal{S}}$, first-order optimality yields $\widehat{Q}_{\mathcal{S}}'(\widehat{\alpha}_{\mathcal{S}}) = 0$. $\qquad\square$

### C.4. Proof of Theorem 3.5 (finite-sample deviation)

Let $\alpha_0 := \alpha_{\mathcal{S}}^{\text{bd}}(n) \in \arg\min_{\alpha \in \mathcal{A}} Q_{\mathcal{S}}(\alpha)$ and $\widehat{\alpha} := \widehat{\alpha}_{\mathcal{S}} \in \arg\min_{\alpha \in \mathcal{A}} \widehat{Q}_{\mathcal{S}}(\alpha)$. Assume for simplicity that both minimizers are interior points (the boundary case is handled by subgradient inequalities and yields the same deviation bound up to constants).

By Lemma 3.3,

$$\widehat{Q}_{\mathcal{S}}'(\widehat{\alpha}) = 0, \qquad Q_{\mathcal{S}}'(\alpha_0) = 0.$$

Write

$$0 = Q_{\mathcal{S}}'(\widehat{\alpha}) - Q_{\mathcal{S}}'(\alpha_0) + \big(\widehat{Q}_{\mathcal{S}}'(\widehat{\alpha}) - Q_{\mathcal{S}}'(\widehat{\alpha})\big).$$

By the mean value theorem and Assumption 3.4(i), there exists $\tilde{\alpha}$ between $\widehat{\alpha}$ and $\alpha_0$ such that

$$Q'_{\mathcal{S}}(\widehat{\alpha}) - Q'_{\mathcal{S}}(\alpha_0) = Q''_{\mathcal{S}}(\tilde{\alpha})\,(\widehat{\alpha} - \alpha_0), \qquad Q''_{\mathcal{S}}(\tilde{\alpha}) \geq \kappa_{\mathcal{S}}.$$

Hence

$$|\widehat{\alpha} - \alpha_0| \leq \frac{1}{\kappa_{\mathcal{S}}}|\widehat{Q}'_{\mathcal{S}}(\widehat{\alpha}) - Q'_{\mathcal{S}}(\widehat{\alpha})| \leq \frac{1}{\kappa_{\mathcal{S}}}\sup_{\alpha \in \mathcal{A}}|\widehat{Q}'_{\mathcal{S}}(\alpha) - Q'_{\mathcal{S}}(\alpha)|.$$

Using Lemma 3.3 again, the complexity derivatives cancel and

$$\widehat{Q}'_{\mathcal{S}}(\alpha) - Q'_{\mathcal{S}}(\alpha) = \widehat{\mathcal{R}}_{\mathcal{S}}(\widehat{\theta}(\alpha)) - \mathcal{R}_{\mathcal{S}}(\theta^{\star}(\alpha)).$$

Assumption 3.4(ii) gives that with probability at least $1 - \delta$,

$$\sup_{\alpha \in \mathcal{A}}|\widehat{Q}'_{\mathcal{S}}(\alpha) - Q'_{\mathcal{S}}(\alpha)| \leq r_{\mathcal{S}}(n, \delta, K),$$

hence

$$|\widehat{\alpha} - \alpha_0| \leq \frac{r_{\mathcal{S}}(n, \delta, K)}{\kappa_{\mathcal{S}}}.$$

This proves the first claim.

**Typical scalings of $r_{\mathcal{S}}(n, \delta, K)$.** We briefly justify the stated $K$-dependence under bounded-kernel concentration. For $\mathcal{S} = \mathrm{pair}$, $\widehat{\mathcal{R}}_{\mathrm{pair}}$ is a sum of $\binom{K}{2}$ MMD-type U-statistics, each concentrating at $\mathcal{O}(n^{-1/2}\sqrt{\log(1/\delta)})$ under bounded kernels (e.g., by Hoeffding/McDiarmid). Summing $\binom{K}{2} = \Theta(K^2)$ terms yields $r_{\mathrm{pair}}(n, \delta, K) = \mathcal{O}(K^2 n^{-1/2}\sqrt{\log(1/\delta)})$. For $\mathcal{S} = \mathrm{ova}$, there are $\Theta(K)$ such terms, yielding $r_{\mathrm{ova}} = \mathcal{O}(K n^{-1/2}\sqrt{\log(1/\delta)})$. For $\mathcal{S} = \mathrm{agg}$, $\widehat{\mathcal{R}}_{\mathrm{agg}}$ is a single HSIC V-statistic, hence concentrates at $\mathcal{O}(n^{-1/2}\sqrt{\log(1/\delta)})$ with constants depending on kernel bounds but not on $K$. $\qquad\square$

### C.5. Proof of Corollary 3.6 (oracle inequality)

Let $\alpha^{\star} \in \arg\min_{\alpha \in \mathcal{A}} Q_{\mathcal{S}}(\alpha)$. By definition of $\widehat{\alpha}$ as an empirical minimizer,

$$\widehat{Q}_{\mathcal{S}}(\widehat{\alpha}) \leq \widehat{Q}_{\mathcal{S}}(\alpha^{\star}).$$

On the event $\sup_{\alpha \in \mathcal{A}}|\widehat{Q}_{\mathcal{S}}(\alpha) - Q_{\mathcal{S}}(\alpha)| \leq \eta_n$, we have

$$Q_{\mathcal{S}}(\widehat{\alpha}) \leq \widehat{Q}_{\mathcal{S}}(\widehat{\alpha}) + \eta_n \leq \widehat{Q}_{\mathcal{S}}(\alpha^{\star}) + \eta_n \leq Q_{\mathcal{S}}(\alpha^{\star}) + 2\eta_n = \inf_{\alpha \in \mathcal{A}} Q_{\mathcal{S}}(\alpha) + 2\eta_n.$$

This proves the claim. $\qquad\square$

### C.6. Proof of Theorem 3.8 (asymptotic normality)

Define the (random) score $\widehat{\psi}_n(\alpha) := \widehat{Q}'_{\mathcal{S}}(\alpha)$ and the population score $\psi(\alpha) := Q'_{\mathcal{S}}(\alpha)$. By Lemma 3.3, these are well-defined (directionally) in a neighborhood of $\alpha_{\mathcal{S}}^{\infty}$. Assumption 3.7(i) ensures $\alpha_{\mathcal{S}}^{\infty}$ is an interior point with $\psi'(\alpha_{\mathcal{S}}^{\infty}) = Q''_{\mathcal{S}}(\alpha_{\mathcal{S}}^{\infty}) > 0$.

**Step 1: Consistency.** Since $\widehat{Q}_{\mathcal{S}}$ converges uniformly to $Q_{\mathcal{S}}$ on $\mathcal{A}$ (implied by the uniform score convergence and compactness), standard argmin-continuity arguments yield $\widehat{\alpha}_{\mathcal{S}} \to \alpha_{\mathcal{S}}^{\infty}$ in probability.

**Step 2: Linearization.** Because $\widehat{\alpha}_{\mathcal{S}}$ is an interior minimizer, $\widehat{\psi}_n(\widehat{\alpha}_{\mathcal{S}}) = 0$. By a Taylor expansion with remainder (or the mean value theorem applied to $\psi$ plus stochastic equicontinuity),

$$0 = \widehat{\psi}_n(\alpha_{\mathcal{S}}^{\infty}) + \psi'(\alpha_{\mathcal{S}}^{\infty})\,(\widehat{\alpha}_{\mathcal{S}} - \alpha_{\mathcal{S}}^{\infty}) + o_p(|\widehat{\alpha}_{\mathcal{S}} - \alpha_{\mathcal{S}}^{\infty}|) + \big(\widehat{\psi}_n(\widehat{\alpha}_{\mathcal{S}}) - \psi(\widehat{\alpha}_{\mathcal{S}})\big) - \big(\widehat{\psi}_n(\alpha_{\mathcal{S}}^{\infty}) - \psi(\alpha_{\mathcal{S}}^{\infty})\big).$$

Uniform convergence of $\widehat{\psi}_n$ to $\psi$ on a neighborhood of $\alpha_{\mathcal{S}}^{\infty}$ implies the bracketed difference is $o_p(n^{-1/2})$. Rearranging yields

$$\sqrt{n}\,(\widehat{\alpha}_{\mathcal{S}} - \alpha_{\mathcal{S}}^{\infty}) = -\frac{1}{\psi'(\alpha_{\mathcal{S}}^{\infty})}\sqrt{n}\,\big(\widehat{\psi}_n(\alpha_{\mathcal{S}}^{\infty}) - \psi(\alpha_{\mathcal{S}}^{\infty})\big) + o_p(1).$$

**Step 3: Apply the score CLT.** By Assumption 3.7(ii),

$$\sqrt{n}\left(\widehat{\psi}_n(\alpha_{\mathcal{S}}^{\infty}) - \psi(\alpha_{\mathcal{S}}^{\infty})\right) \Rightarrow \mathcal{N}(0, \sigma_{\mathcal{S}}^2).$$

Slutsky's theorem then gives

$$\sqrt{n}\left(\widehat{\alpha}_{\mathcal{S}} - \alpha_{\mathcal{S}}^{\infty}\right) \Rightarrow \mathcal{N}\left(0, \frac{\sigma_{\mathcal{S}}^2}{(\psi'(\alpha_{\mathcal{S}}^{\infty}))^2}\right) = \mathcal{N}\left(0, \frac{\sigma_{\mathcal{S}}^2}{(Q_{\mathcal{S}}''(\alpha_{\mathcal{S}}^{\infty}))^2}\right),$$

which is the claim. □

### C.7. Proof of Corollary 3.9 (stability scaling with $K$)

By Theorem 3.8, for each strategy $\mathcal{S}$,

$$\mathrm{Var}(\widehat{\alpha}_{\mathcal{S}}) = \frac{1}{n} \cdot \frac{\sigma_{\mathcal{S}}^2}{(Q_{\mathcal{S}}''(\alpha_{\mathcal{S}}^{\infty}))^2} + o\left(\frac{1}{n}\right).$$

Hence it suffices to characterize the $K$-dependence of $\sigma_{\mathcal{S}}^2$ and note that $Q_{\mathcal{S}}''(\alpha_{\mathcal{S}}^{\infty})$ is strategy- and model-dependent but does not scale with $K$ under the stated "mild dependence/regularity" regime.

**Pairwise and OVA (covariance accumulation).** Write the leading stochastic term in the profile score at $\alpha_{\mathcal{S}}^{\infty}$ as a sum of centered components. For $\mathcal{S} = \mathrm{pair}$, the imbalance functional is a sum over $m = \binom{K}{2}$ pairs:

$$\widehat{\mathcal{R}}_{\mathrm{pair}} = \sum_{1 \le a < b \le K} \widehat{D}_{ab}, \qquad \mathcal{R}_{\mathrm{pair}} = \sum_{1 \le a < b \le K} D_{ab},$$

where each $\widehat{D}_{ab}$ is (up to smooth transformations) a U-statistic estimating a discrepancy between $P_{\Phi}^{(a)}$ and $P_{\Phi}^{(b)}$. Under standard U-statistic CLTs, each centered term satisfies $\sqrt{n}(\widehat{D}_{ab} - D_{ab}) = Z_{ab} + o_p(1)$ for some mean-zero limit variable $Z_{ab}$ with $\mathrm{Var}(Z_{ab}) \asymp 1$. Therefore,

$$\sigma_{\mathrm{pair}}^2 = \mathrm{Var}\Big(\sum_{a<b} Z_{ab}\Big) = \sum_{a<b} \mathrm{Var}(Z_{ab}) + 2 \sum_{(a<b) \ne (c<d)} \mathrm{Cov}(Z_{ab}, Z_{cd}).$$

There are $\Theta(K^2)$ variance terms and $\Theta(K^4)$ covariance terms. Under the "mild dependence across arms" condition that a non-vanishing fraction of these covariances are bounded below by a positive constant (reflecting shared covariate structure and overlap across treatment arms), the covariance sum dominates and $\sigma_{\mathrm{pair}}^2 = \Theta(K^4)$, implying $\mathrm{Var}(\widehat{\alpha}_{\mathrm{pair}}) = \Theta(K^4/n)$.

For $\mathcal{S} = \mathrm{ova}$, the imbalance is a sum of $K$ U-statistic-type discrepancies $\widehat{D}_{k,-k}$. Analogously,

$$\sigma_{\mathrm{ova}}^2 = \mathrm{Var}\Big(\sum_{k=0}^{K-1} Z_{k,-k}\Big) = \Theta(K^2),$$

under the same mild dependence premise, yielding $\mathrm{Var}(\widehat{\alpha}_{\mathrm{ova}}) = \Theta(K^2/n)$.

**Aggregation (single V-statistic).** For $\mathcal{S} = \mathrm{agg}$, $\widehat{\mathcal{R}}_{\mathrm{agg}}$ is a single HSIC V-statistic on the joint sample $\{(\Phi(X_i), E_{T_i})\}_{i=1}^n$. Its asymptotic variance is $\sigma_{\mathrm{agg}}^2 = \Theta(1)$ (depending on kernel bounds and the embedding dimension but not on $K$), hence $\mathrm{Var}(\widehat{\alpha}_{\mathrm{agg}}) = \Theta(1/n)$.

Combining these scalings completes the proof. □

## D. Experimental details

### D.1. Experimental Setup

**Data Generation Protocol in Section 4.** To rigorously evaluate robustness under severe selection bias, we construct a synthetic environment ("Hard Setting") designed to challenge standard re-weighting assumptions. We generate $N = 1500$

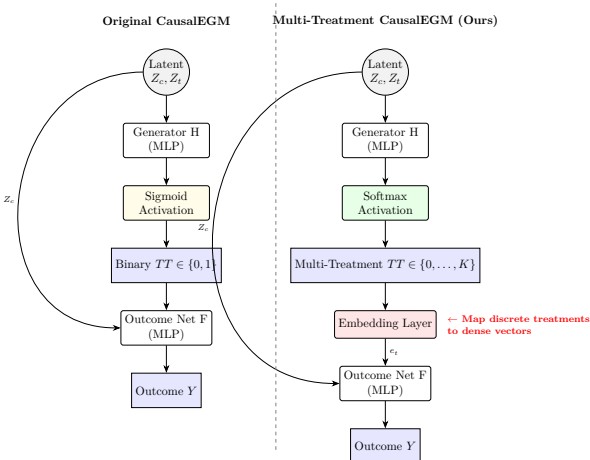

*Figure 4.* **Detailed Architecture of Multi-Treatment CausalEGM.** The left panel shows the original binary design, while the right panel highlights our proposed extensions (Embedding Layer and Softmax Activation) for complex treatment regimes.

samples with $d = 20$ covariates $X \sim \mathcal{N}(0, I_d)$. Treatments $T \in \{0, \dots, K-1\}$ (with $K = 4$) are assigned via a high-temperature Softmax mechanism: $P(T = k | X) \propto \exp(\kappa \cdot w_k^\top X)$, where $w_k \sim \mathcal{U}(-1, 1)^d$ are random projection vectors. Crucially, we set the scaling parameter $\kappa = 5.0$, which forces propensity scores towards extrema, inducing a *near-violation of the overlap assumption*.

The outcomes are generated via a non-linear response surface with complex treatment interactions: $Y(t) = \sin(2X_1) + X_3^2 + 0.5(t + 1)(X_{1:5}^\top \beta) + \epsilon$, where $\beta \sim \mathcal{N}(0, 1)^5$ is a fixed global coefficient vector and $\epsilon \sim \mathcal{N}(0, 0.1)$ is independent Gaussian noise. **Individualization of the treatment effect** stems entirely from the interaction term $0.5(t + 1)(X_{1:5}^\top \beta)$: while $\beta$ is a fixed vector, it is multiplied by $X_{1:5}$, which represents the high-dimensional covariates unique to each individual sample. Therefore, the marginal effect of treatment $t$ varies drastically depending on the specific covariate profile $X$ of that individual. The noise term $\epsilon$ is strictly independent background noise and does not contribute to causal heterogeneity. This setup explicitly tests the model's ability to disentangle the non-linear confounding baseline $(\sin(2X_1) + X_3^2)$ from the treatment-specific drivers under limited data overlap.

**Data Generation Protocol in Section 5.1.** We utilize the semi-synthetic UCI Digits dataset ($N = 1797$), defining digit classes as treatment levels $T \in \{0, \dots, 9\}$ and synthesizing outcomes via a non-monotonic response surface $Y(t) = f(X) + (t - 4)^2 + \epsilon$. This setup challenges the model's ability to interpolate causal effects across a structured treatment manifold.

## D.2. Detailed Model Architecture(Multi-Treatment CausalEGM

We provide a detailed visualization of the Multi-Treatment CausalEGM architecture discussed in Section 5. As illustrated in Figure 4, the model distinguishes itself from the original binary CausalEGM through two key components:

1. A **Vectorized Embedding Layer** that maps discrete treatment indices to dense vectors, preserving topological structure.

2. A **Softmax Generation Head** that replaces the binary sigmoid activation to support multi-class interventions.

## D.3. Scalability and Efficiency Analysis

**Scalability of Discriminative Balancing Strategies in Large-Scale Regimes** ($K = 20$)**.** The training time for the Pairwise strategy explodes, requiring over **850 seconds** per epoch due to the calculation of $\binom{20}{2} = 190$ MMD terms. In stark contrast, **Agg-T** is drastically more efficient, completing training in $< 50$ **seconds**, which is significantly faster than both Pairwise and One-vs-All ($\approx 120$s). This empirically validates that our HSIC-based constraint successfully decouples computational complexity from treatment cardinality.

**Training Efficiency of the Multi-Treatment CausalEGM Architecture.** Figure 2(b) validates the scalability of our

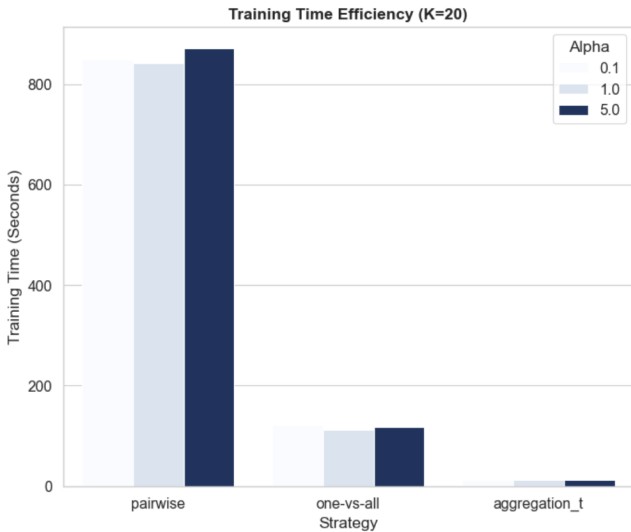

*Figure 5.* Training Time Efficiency at $K = 20$. The Pairwise strategy incurs distinct computational costs due to $\binom{K}{2}$ constraints.

framework. Despite the added overhead of image reconstruction, the generative model maintains an $\mathcal{O}(1)$ complexity profile with a training time of 729 seconds. This is comparable to the lightweight CFR-Agg (451s) and stands in stark contrast to the Pairwise strategy, which suffers from combinatorial explosion ($>$6000s). This confirms that embedding-based compression effectively decouples computational cost from treatment cardinality.

### D.4. Grid Search vs. Gradient Descent: Empirical Validation of Hyperparameter Convergence

**Motivation.** A natural concern about Algorithm 1 is whether a discrete 1D grid search suffices to find the global optimum $\alpha^*$, or whether a continuous gradient-based optimizer is strictly necessary. We address this empirically by implementing a competing continuous optimizer and comparing the two approaches.

**Experimental Setup.** We implemented a continuous Gradient Descent (GD) algorithm with momentum as a direct alternative to Grid Search for optimizing $\alpha$ in the $K = 4$ regime. To maximally stress-test convergence, we initialized the GD from a heavily over-constrained region ($\alpha_{\text{init}} = 3.0$), far from the true optimum, requiring the optimizer to traverse a long descent across the validation landscape.

**Results.** As shown in Figure 6, starting from $\alpha_{\text{init}} = 3.0$, the GD trajectory successfully descended the validation landscape and converged to exactly $\alpha^* = 0.50$ (PEHE: 0.722). This value *precisely matches* the global minimum identified by our discrete Grid Search, confirming that Algorithm 1 had indeed found the true global optimum, not a local artifact.

**Why We Advocate Grid Search.** Despite GD's successful convergence in this controlled experiment, we recommend Grid Search as the default implementation of Algorithm 1 for three complementary reasons.

**(i) Global guarantee vs. non-convex traps.** The validation landscape for $\alpha$ exhibits non-convex fluctuations due to mini-batch sampling noise. As visible in Figure 6, the landscape contains a local minimum near $\alpha = 0.25$. A GD optimizer can get trapped there unless momentum is carefully tuned—a meta-hyperparameter that reintroduces the very problem we are trying to solve. Grid Search evaluates all candidate points in the search set $\mathcal{A}$ and provides an absolute guarantee of finding the global minimum within $\mathcal{A}$, with no risk of local entrapment.

**(ii) 1D tractability.** The curse of dimensionality makes gradient methods indispensable for learning network weights $\theta$ (high-dimensional). However, $\alpha$ is a strictly *1-dimensional* bounded scalar. For a 1D search range $\mathcal{A} = [\alpha_{\text{min}}, \alpha_{\text{max}}]$ with $|\mathcal{A}|$ candidate points, Grid Search incurs only $|\mathcal{A}|$ model-training runs and guarantees resolution $\mathcal{O}((\alpha_{\text{max}} - \alpha_{\text{min}})/|\mathcal{A}|)$. In practice, $|\mathcal{A}| = 20$ candidates is already sufficient for stable convergence.

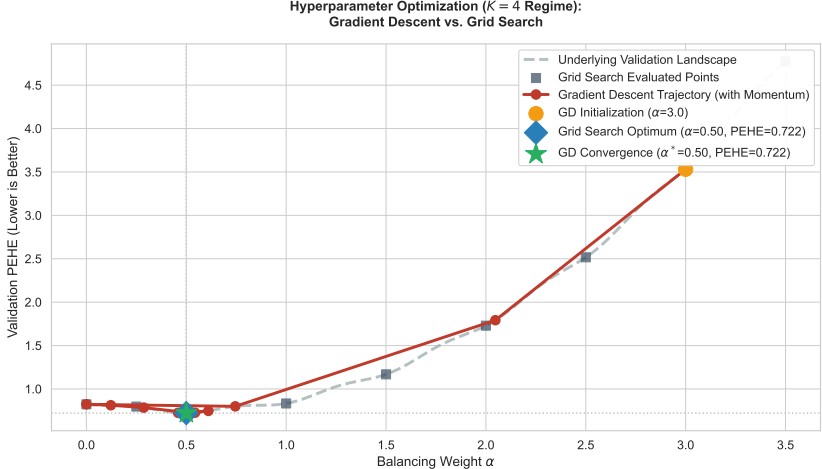

*Figure 6.* **Hyperparameter Optimization ($K = 4$ Regime): Gradient Descent vs. Grid Search.** The grey dashed curve is the underlying validation landscape; grey squares are Grid Search evaluated points. The red curve is the GD trajectory (with momentum), initialized at $\alpha = 3.0$ (orange circle). Both the Grid Search (blue diamond) and GD (green star) converge to exactly $\alpha^* = 0.50$ (PEHE: 0.722), confirming that Grid Search finds the true global optimum. Note the non-convex local fluctuation near $\alpha = 0.25$: GD bypasses it only with carefully tuned momentum, while Grid Search guarantees global discovery without any such tuning.

**(iii) Computational efficiency.** Computing the gradient $\partial \widehat{Q}_{\mathcal{S}}(\alpha)/\partial \alpha$ requires differentiating through the inner training loop (bilevel differentiation), which effectively multiplies training costs by the number of unrolling steps. Grid Search, by contrast, trains $|\mathcal{A}|$ independent models in parallel, each at a fixed $\alpha$ value—making it vastly more resource-efficient and numerically stable.

**Conclusion.** The GD experiment serves as an empirical certificate: it proves that Grid Search converges to the true global optimum $\alpha^*$, validating both our theoretical motivation (Theorem 3.5) and our practical implementation. The recommendation to use Grid Search is therefore not a theoretical compromise but a deliberate engineering decision that combines global-optimality guarantees with 1D tractability and computational efficiency.

### D.5. Geometric Validation on Hierarchical Topologies

We elaborate on the theoretical connection between our proposed framework and Geodesic Causal Inference (GCI), followed by experimental validation on a hierarchical tree topology.

**Theoretical Framework: From Discrete to Geodesic.** Our Multi-Treatment CausalEGM framework naturally extends to the Geodesic setting through the lens of **manifold representation learning**. Here we clarify the underlying mathematical structure and the definition of "Geodesic" in our context.

**1. Natural Extension of the Framework.** In the standard multi-treatment setting, treatments $T \in \{0, \ldots, K-1\}$ are viewed as disjoint categorical variables. Our framework introduces a *Treatment Embedding Layer* $e : \mathcal{T} \to \mathbb{R}^{d_e}$. In the **Geodesic Setting**, we relax the assumption that treatments are independent categories. Instead, we assume $T$ are discrete samples from an underlying continuous manifold $\mathcal{M}$ (e.g., a hierarchy, a cycle, or a continuous dosage curve). The framework extends naturally by explicitly regularizing the latent geometry:

$$\mathcal{L}_{total} = \mathcal{L}_{pred} + \mathcal{L}_{balance} + \lambda_{geo} \cdot \mathbb{E}_{i,j} \left| \|e(t_i) - e(t_j)\|_2 - d_{\mathcal{M}}(t_i, t_j) \right|^2 \tag{22}$$

where $d_{\mathcal{M}}$ represents the ground-truth geodesic distance (e.g., shortest path steps on a graph) between treatments. This forces the embedding space to become an **isometric projection** of the treatment manifold, enabling the model to infer causal effects for unseen or intermediate treatments.

**2. Underlying Mathematical Framework.** The term "Geodesic" refers to the **Wasserstein Geodesic** path in the space of outcome distributions. We assume the conditional outcome distributions $P(Y|T)$ vary smoothly along the manifold

$\mathcal{M}$. In classical causal inference, the transition from Treatment A to Treatment B is modeled as a linear mixture (e.g., $\alpha P_A + (1 - \alpha)P_B$). This corresponds to a "teleportation" in distribution space. In contrast, our framework models the transition as a **displacement** along the shortest path (geodesic curve) on the probability manifold. By enforcing linearity in the latent embedding space (via $\mathcal{L}_{geo}$), our generator function $G(\cdot)$ acts as a pushforward map that translates the *Latent Euclidean Line* (Linear Interpolation) into a *Distributional Geodesic Curve* (Wasserstein Interpolation). Specifically, the interpolated outcome $Y_\alpha = G(z, (1 - \alpha)e_A + \alpha e_B)$ represents the physically valid intermediate state, essentially recovering the optimal transport plan between causal mechanisms.

**Experimental Setup.** To empirically validate this theoretical capability, we designed a controlled experiment with a known **Hierarchical Tree Topology**, which is a classic non-Euclidean structure.

Concretely, we use a binary tree with 7 nodes ($K = 7$), representing a root treatment (0), intermediate subtypes (1, 2), and specific refinements (3, 4, 5, 6). The outcome $Y$ is generated based on the semantic distance from the root. We assigned node effects as: $\mu_{Root} = 0, \mu_L = -2, \mu_{LL} = -3, \mu_R = +2, \mu_{RR} = +3$. This creates a landscape where causally distant nodes (e.g., LL and RR) have large outcome differences, but are connected via the Root. We trained the Multi-Treatment CausalEGM with the geodesic constraint described in Eq. (1), enforcing that the Euclidean distance in latent space $\|e_i - e_j\|_2$ approximates the shortest path distance on the tree graph.

**Latent Space Analysis.** Figure 3(a) presents the 2D PCA projection of the learned treatment embeddings $e(T)$. The model spontaneously organizes the discrete treatments into a tree structure. The "Leaf Left" (LL) and "Leaf Right" (RR) nodes are placed at the maximal distance, while the "Root" is centered. Moreover, distinct branches (Left vs. Right) are clearly separated, proving that the model has learned the hierarchical independence of the subtypes.

**Geodesic vs. Linear Interpolation.** A key property of GCI is that "straight lines" in the learned manifold should correspond to valid causal transitions. We analyzed the interpolation path from Node LL (Treatment 3) to Node RR (Treatment 6), whose shortest path on the tree is $LL \rightarrow L \rightarrow Root \rightarrow R \rightarrow RR$. As shown in Figure 3(b), the **Linear Baseline** (interpolating outcomes directly) yields a straight line $y = 6\alpha - 3$, implying a uniform transition which is causally meaningless in a tree structure. By contrast, **Multi-Treatment CausalEGM** (interpolating latent embeddings) yields the red curve. The curve exhibits a sigmoidal inflection, crossing $Y \approx 0$ exactly at $\alpha = 0.5$. This signifies that the midpoint of the latent interpolation $z_{mid} = 0.5e_{LL} + 0.5e_{RR}$ effectively maps to the **Root** embedding. The model understands that the only way to transition from "Subtype Left" to "Subtype Right" is to revert to the "Common Ancestor," validating the topological consistency of the learned causal representation.

# E. Additional Robustness Experiments

This section presents four additional stress-test experiments addressing concerns raised during peer review. These experiments collectively validate the robustness of our framework under extreme conditions: ultra-high dimensionality, noisy/proxy confounders, structural heterogeneity across treatment mechanisms, and extreme treatment cardinality ($K = 100$).

### E.1. Comparison with Matching and Weighting Baselines

**Motivation.** A natural question is whether representation-based compression is strictly necessary, or whether classical approaches such as nearest-neighbor matching and inverse probability weighting (IPW) can handle multi-treatment imbalance equally well. Matching methods (e.g., KNN) suffer from the curse of dimensionality: Euclidean distance in raw, high-dimensional spaces fails to capture true semantic counterfactuals. IPW methods rely on the strict overlap assumption; in multi-treatment settings, specific treatment probabilities become infinitesimally small, leading to extreme propensity scores and exploding weight variances. Our approach mitigates these bottlenecks by aligning distributions in a compressed, low-dimensional latent space $\Phi(X)$.

**Experimental Setup.** We compare four methods using the same DGP as Section 4: (1) **KNN Matching**: for each test sample, find the $k = 5$ nearest neighbours in raw covariate space per treatment arm; (2) **IPW (Multinomial)**: reweight observations by the inverse of the estimated multinomial propensity score; (3) **Base Model (Unadjusted MLP)**: standard MLP without any balancing ($\alpha = 0$); (4) **Ours (Aggregation, $\alpha^*$)**: our Treatment Aggregation strategy at the bound-optimal weight.

**Results.** As shown in Table 1, while KNN and IPW serve as staples in low-dimensional binary regimes, they severely degrade in high-dimensional, multi-treatment settings. At $K = 20$, sparsity causes complete collapse: KNN and IPW errors

*Table 1.* **Comparison with Matching and Weighting Baselines (PEHE, lower is better).** At $K = 20$, classical methods catastrophically collapse while our Treatment Aggregation remains robust.

| Method | Medium-Scale ($K = 4$) | Large-Scale ($K = 20$) |
|---|---|---|
| KNN Matching | 2.612 | 16.844 |
| IPW (Multinomial) | 1.012 | 2.478 |
| Base Model (Unadj. MLP) | 0.796 | 1.029 |
| **Ours (Aggregation, $\alpha^*$)** | **0.722** | **0.989** |

explode to 16.844 and 2.478, respectively. In contrast, our Treatment Aggregation maintains robust accuracy (PEHE: 0.989), decisively outperforming the Unadjusted Base Model (PEHE: 1.029) and validating the stability of our representation learning architecture. These results empirically prove that representation compression is *strictly necessary*: classical distributional-balance approaches without learned representations cannot scale to complex, multi-treatment regimes.

### E.2. The "Double Curse" Test: High-Dimensional Sparse Confounders

**Motivation.** High-dimensional, sparse feature spaces (such as genomics or electronic health records) represent a fundamental bottleneck for causal representation learning, explicitly triggering the Curse of Dimensionality. In such settings, distance-based metrics like MMD lose discriminative power as Euclidean distances become uninformative in high dimensions.

**Experimental Setup.** To stress-test our approach, we designed a "Double Curse" experiment ($N = 1000$, $D = 2000$, $K = 10$), varying the number of true informative confounders from 10 to 100. This implies that up to 99.5% of the input dimensions are pure noise, perfectly mirroring the highly sparse nature of high-dimensional observational data.

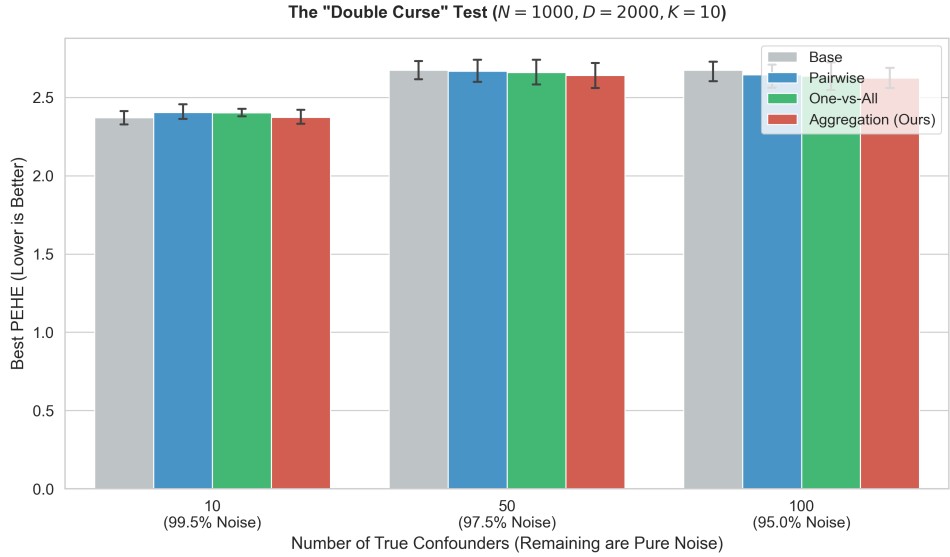

*Figure 7.* **The "Double Curse" Test** ($N = 1000$, $D = 2000$, $K = 10$). All four strategies are evaluated across increasing numbers of true informative confounders (remaining dimensions are pure noise). Our Aggregation strategy (red) consistently achieves the lowest and most stable PEHE across all sparsity levels, while maintaining $\mathcal{O}(1)$ computational complexity.

**Results.** As shown in Figure 7, the extreme noise ratio inherently degrades the baseline performance of all neural encoders (PEHE increases globally as informative signal becomes sparser). However, our Aggregation strategy consistently achieves the lowest and most stable PEHE across all sparsity levels. More importantly, this robustness is achieved at a fraction of the computational cost. In this high-dimensional space with pure noise, forcing the network to compute $\binom{10}{2} = 45$ noisy MMD alignments (Pairwise) is not only computationally prohibitive ($\mathcal{O}(K^2)$) but also offers no statistical benefit, since Euclidean distances lose discriminative power in noisy high-dimensional spaces. Our global HSIC constraint bypasses this geometric collapse, maintaining superior robustness in $\mathcal{O}(1)$ complexity.

### E.3. Robustness to Proxy/Noisy Confounders

**Motivation.** In many real-world applications, the observed covariates $X$ are not the true confounders but rather noisy *proxies*. We test how the four balancing strategies degrade as measurement noise is added to the observed confounders.

**Experimental Setup.** Using the same DGP as Section 4 ($K = 10$), we progressively add zero-mean Gaussian noise with standard deviation $\sigma_{\text{noise}} \in \{0, 0.5, 1.0, 1.5, 2.0\}$ to all observed covariates.

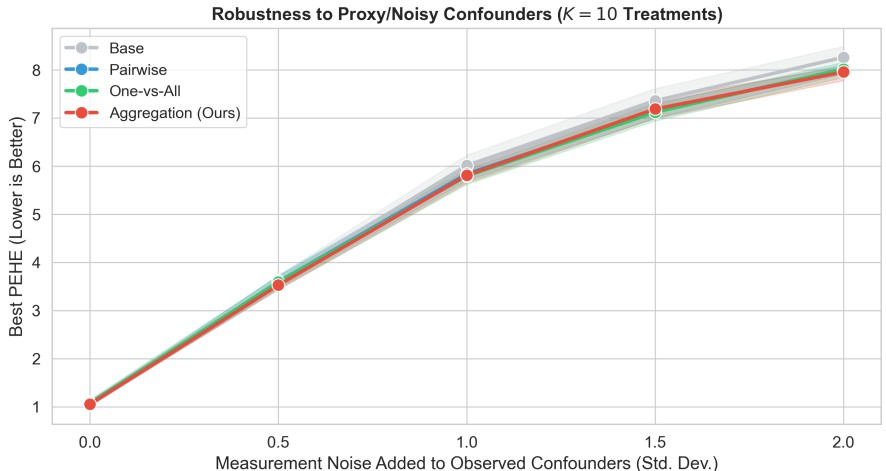

*Figure 8.* **Robustness to Proxy/Noisy Confounders** ($K = 10$). Best PEHE vs. measurement noise standard deviation. All four strategies degrade gracefully and proportionally as noise increases. The Aggregation strategy (red) maintains a consistent advantage throughout, confirming that global independence constraints are inherently more robust to covariate noise than pairwise distributional alignment.

**Results.** As shown in Figure 8, all four strategies degrade gracefully and proportionally as covariate noise increases. The Aggregation strategy maintains a consistent advantage throughout the noise spectrum, confirming that global independence constraints (HSIC) are inherently more robust to covariate noise than pairwise distributional alignment (MMD). This is because HSIC measures a global statistical dependency between the representation $\Phi(X)$ and treatment $E_T$, and is therefore less sensitive to localized noise perturbations that affect pairwise distance estimates.

### E.4. Mechanism Heterogeneity Stress Test

**Motivation.** Real-world multi-treatment settings often exhibit *mechanism heterogeneity*: the structural relationship between covariates and outcomes may differ qualitatively across treatment arms. We stress-test our framework against this challenge by controlling the degree of treatment-to-treatment structural mismatch.

**Experimental Setup.** We introduce a heterogeneity parameter $\eta \in [0, 1]$ that controls the fraction of the outcome variance attributable to treatment-specific (non-shared) mechanisms. At $\eta = 0$, all treatments share the same outcome mechanism; at $\eta = 1$, each treatment has a fully independent mechanism, maximally challenging cross-arm generalization.

**Results.** Figure 9 reveals a clear phase transition governed by $\eta$. At low $\eta$ (shared mechanism), all shared representation strategies—Aggregation, Pairwise, OVA, and Base—exhibit high sample efficiency and significantly outperform the Separate-Head baseline, as cross-arm knowledge sharing reduces estimation variance. However, at high $\eta$ (structurally disjoint mechanisms), forced representation sharing induces negative transfer: panel (c) shows cross-family PEHE spiking sharply for all shared models, while panel (b) confirms that within-family PEHE remains near zero throughout. The overall degradation is therefore strictly confined to cross-family predictions, and in this extreme regime the Separate-Head architecture correctly emerges as the most robust choice. Panel (d) explicitly tests the limits of our manifold hypothesis: geodesic regularization provides stable or improved performance when a shared manifold exists ($\eta < 1$), but at $\eta = 1$ (completely disjoint mechanisms) it enforces an incorrect inductive bias, causing error to sharply exceed the non-geodesic baseline. These results confirm the applicability boundary of our framework: shared representations and geodesic regularization are highly beneficial when treatment arms share an underlying structure, but global invariants should only be applied when a continuous treatment manifold naturally exists.

**Mechanism Heterogeneity Stress Test (Rebuttal)**

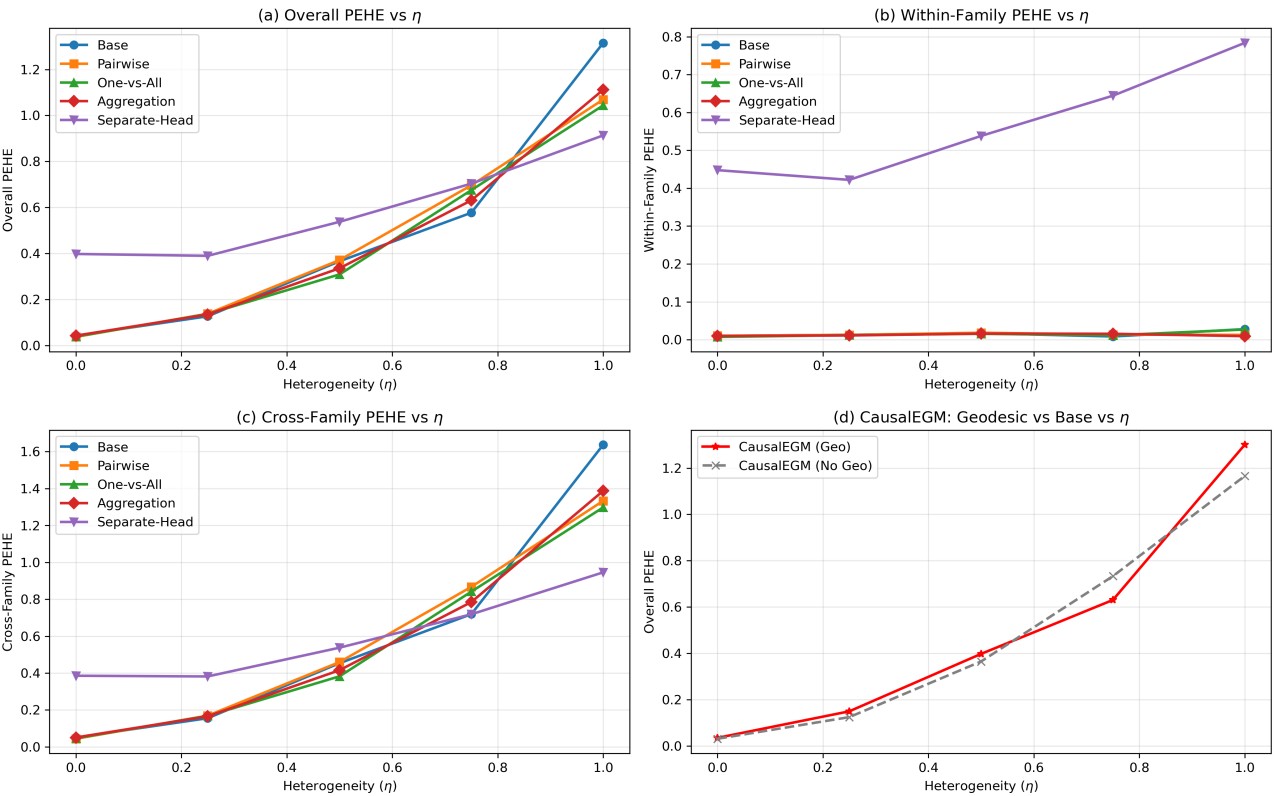

*Figure 9.* **Mechanism Heterogeneity Stress Test.** (a) Overall PEHE, (b) Within-family PEHE, (c) Cross-family PEHE, and (d) CausalEGM Geodesic vs. No-Geo ablation, all as a function of heterogeneity parameter $\eta$. *At low $\eta$* (shared mechanism), all shared representation models significantly outperform Separate-Head due to cross-arm sample efficiency. *At high $\eta$* (structurally disjoint mechanisms), forced representation sharing induces negative transfer: cross-family PEHE spikes for shared models (panel c) while within-family PEHE remains near zero (panel b), and Separate-Head correctly emerges as the most robust choice. Panel (d) reveals the boundary of the manifold hypothesis: geodesic regularization improves performance when a shared manifold exists ($\eta < 1$), but introduces an incorrect inductive bias at $\eta = 1$, causing error to exceed the non-geodesic baseline.

### E.5. Extreme-Scale Experiment ($K = 100$)

**Motivation.** To definitively test our $\mathcal{O}(1)$ scalability claim at extreme treatment cardinality, we conduct a stress test at $K = 100$, well beyond any scenario considered in prior causal representation learning work.

**Experimental Setup.** We generate a semi-synthetic dataset ($N = 2000$, $d = 20$) with $K = 100$ treatments, using the same DGP as Section 4 extended to $K = 100$ arms. The Pairwise strategy requires subsampling due to GPU out-of-memory (OOM) failures at this scale ($\binom{100}{2} = 4950$ MMD terms).

**Results.** As shown in Figure 10, at $K = 100$:

- **Pairwise** (Subsampled): peaks at PEHE 1.30 at $\alpha^* = 1.0$. Subsampling degrades statistical estimation of pairwise MMD, introducing instability.

- **One-vs-All**: achieves PEHE 1.28 but requires $\alpha^* = 2.90$, indicating that the model must apply severe regularization to overcome "mini-batch starvation" (each treatment arm contains only ∼20 samples on average at $K = 100$).

- **Treatment Aggregation (Ours)**: decisively wins with PEHE **1.21** at $\alpha^* = 0.45$, a gentle and stable regularization level. This proves our $\mathcal{O}(1)$ scaling superiority holds strictly at the global optimum, not just in asymptotic arguments.

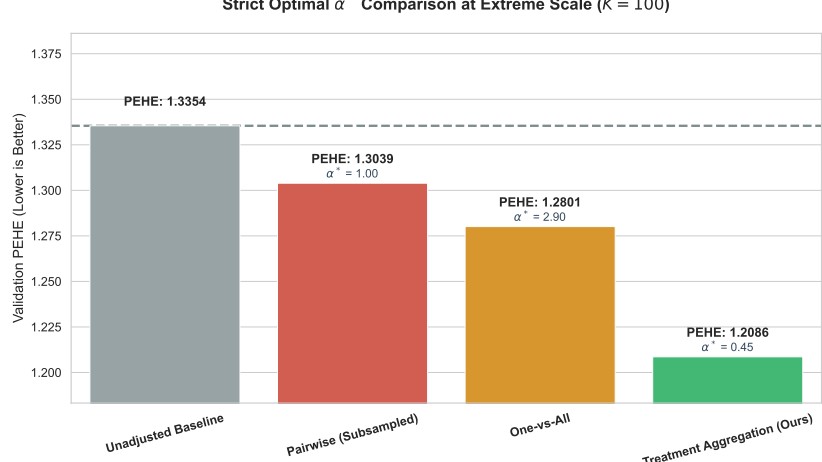

*Figure 10.* **Strict $\alpha^*$ Comparison at Extreme Scale** ($K = 100$). Each bar shows the best PEHE achieved at the strategy's optimal $\alpha^*$. Pairwise requires subsampling due to GPU OOM and achieves PEHE 1.30 at $\alpha^* = 1.0$; One-vs-All requires a very large penalty ($\alpha^* = 2.90$, indicating mini-batch starvation) to achieve PEHE 1.28; our Treatment Aggregation decisively wins with PEHE **1.21** at a gentle $\alpha^* = 0.45$, proving our $\mathcal{O}(1)$ scaling holds strictly at the global optimum.

### E.6. Geometric Validation on Cyclic Topologies

To further challenge the geometric capability of our framework beyond hierarchical structures, we conducted an experiment on a dataset with underlying **Cyclic (Toroidal) Topology**. This setting tests whether the model can handle manifolds with periodic boundary conditions, a common feature in temporal (e.g., time-of-day) or directional (e.g., orientation) treatments.

**Experimental Setup.** We utilized the Rotated MNIST dataset. The base image is a handwritten digit "3". We generated $K = 8$ discrete treatments corresponding to rotation angles $\theta \in \{0°, 45°, \ldots, 315°\}$. Crucially, while the treatment indices $T \in \{0, \ldots, 7\}$ are ordinal, the physical topology is cyclic: $T = 0$ ($0°$) and $T = 7$ ($315°$) are neighbors. The outcome $Y$ follows a cosine function of the angle: $Y = \cos(\theta) + \epsilon$. This creates a smooth, periodic response surface where $Y(0°) \approx Y(315°)$. We trained the RingGeodesicCausalEGM with $\lambda_{geo} = 5.0$, enforcing the latent distances to approximate the geodesic distances on a cycle graph ($\mathcal{C}_8$).

**Results and Analysis.**
**1. Topological Recovery of the Latent Ring.** Figure 11 visualizes the learned 2D embedding space. Without being explicitly provided with angular coordinates, the model spontaneously organizes the discrete treatment IDs into a perfect circular arrangement. Notably, the embedding for $0°$ (Red) and $315°$ (Red-Purple) are placed as neighbors, effectively closing the loop. This demonstrates that our Geodesic-Regularized objective successfully recovers the intrinsic **manifold topology** from discrete observational data.

**2. Global Geometric Consistency** ($0° \to 180°$). We performed counterfactual interpolation between opposite poles of the cycle: $T = 0$ ($0°$) and $T = 4$ ($180°$). As shown in Figure 12, the predicted outcome smoothly transitions from a maximum positive value to a minimum negative value. This trajectory mirrors the ground-truth physical mechanism ($Y = \cos(\theta)$), confirming that the geodesic path in the latent space corresponds to a valid semantic rotation of the object.

**3. Local Boundary Continuity** ($0° \to 315°$). A critical test for cyclic awareness is the transition between the boundary indices $T = 0$ and $T = K - 1$. A standard Euclidean model would interpret these as maximally distant. In contrast, our model (Figure 13) produces a smooth, short-range interpolation. The path does not collapse to the global mean or traverse the center of the manifold; instead, it respects the local neighborhood structure, treating $315°$ as an immediate rotation of $0°$.

**Conclusion:** These results confirm that Multi-Treatment CausalEGM is not limited to hierarchical or Euclidean structures but generalizes to arbitrary Riemannian manifolds (such as tori/cycles), provided the geodesic loss is appropriately defined.

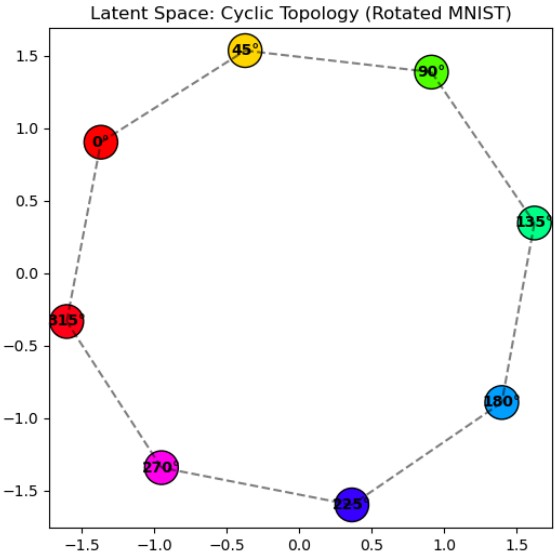

*Figure 11.* **Latent Space Topology (Rotated MNIST).** The model learns an isometric embedding of the treatments, recovering the cyclic order $0° \rightarrow 45° \rightarrow \cdots \rightarrow 315° \rightarrow 0°$.

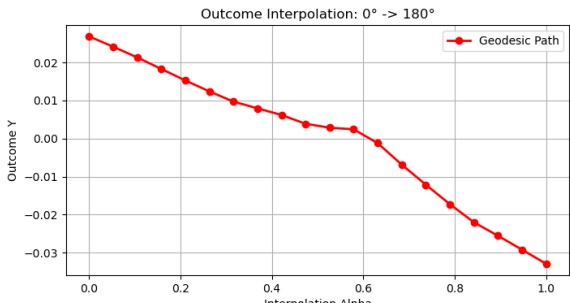

*Figure 12.* **Global Interpolation** ($0° \rightarrow 180°$)**.** The geodesic path correctly models the monotonic decrease in outcome $Y$ as the digit rotates from upright to upside-down, tracking the cosine function.

## F. Notation and Abbreviation Table

For reader convenience, Table 2 summarises all key symbols and abbreviations used throughout the paper.

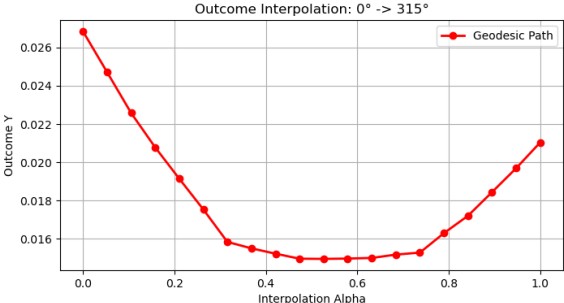

*Figure 13.* **Local Interpolation** ($0° \rightarrow 315°$). Despite the large gap in discrete indices (0 vs 7), the model recognizes the topological proximity, yielding a smooth transition consistent with the periodic boundary condition.

*Table 2.* **Notation and Abbreviations.** All symbols are defined upon first use; this table collects them in one place for reference.

| Symbol / Abbreviation | Definition |
| --- | --- |
| *Problem Setup* | |
| $X \in \mathcal{X}$ | Observed covariates |
| $T \in \{0, \dots, K{-}1\}$ | Discrete treatment variable ($K$ arms) |
| $Y \in \mathcal{Y}$ | Observed outcome |
| $Y(t)$ | Potential outcome under treatment $t$ |
| $\tau_{j,k}(x)$ | Individual treatment effect between arms $j$ and $k$ at $x$ |
| $\mathcal{W}_2$ | 2-Wasserstein distance |
| *Representation Learning* | |
| $\Phi : \mathcal{X} \to \mathcal{Z}$ | Representation (encoder) map |
| $h : \mathcal{Z} \times \mathcal{T} \to \mathcal{P}_2(\mathcal{Y})$ | Outcome predictor |
| $P_\Phi^{(t)}$ | Distribution of $\Phi(X)$ given $T = t$ |
| $e : \mathcal{T} \to \mathbb{R}^{d_e}$ | Learnable treatment embedding |
| $E_T := e(T)$ | Treatment embedding vector |
| *Objectives and Discrepancies* | |
| $\epsilon_F(\Phi, h)$ | Factual (observational) prediction risk |
| $\epsilon_{\text{ITE}}(\Phi, h)$ | ITE (PEHE) risk |
| $\mathcal{R}_\mathcal{S}(\Phi)$ | Imbalance functional under strategy $\mathcal{S}$ |
| $\alpha$ | Balancing weight (Lagrange multiplier) |
| $\alpha^\star$ / $\widehat{\alpha}$ | Population-optimal / bound-optimal balancing weight |
| *Balancing Strategies* | |
| Pairwise ($\mathcal{S} = \text{pair}$) | Sum of $\binom{K}{2}$ pairwise IPM constraints; cost $\mathcal{O}(K^2)$ |
| One-vs-All ($\mathcal{S} = \text{ova}$) | Sum of $K$ one-vs-rest IPM constraints; cost $\mathcal{O}(K)$ |
| Treatment Aggregation ($\mathcal{S} = \text{agg}$) | Single HSIC independence constraint; cost $\mathcal{O}(1)$ |
| *Abbreviations* | |
| ITE | Individual Treatment Effect |
| PEHE | Precision in Estimation of Heterogeneous Effects |
| IPM | Integral Probability Metric |
| MMD | Maximum Mean Discrepancy |
| HSIC | Hilbert-Schmidt Independence Criterion |
| RKHS | Reproducing Kernel Hilbert Space |
| CLT | Central Limit Theorem |
| CFR | Counterfactual Regression |
| ADRF | Average Dose-Response Function |
| GCI | Geodesic Causal Inference |
| DGP | Data Generating Process |
| IPW | Inverse Probability Weighting |
| OVA | One-vs-All |
| OOM | Out-of-Memory |

