# OpenReview forum: "Causal Representation Learning with Optimal Compression and  Complex Treatments"
_ICML.cc/2026/Conference — ICML 2026 regular_

### Official Review · Reviewer_H8X7 · 2026-02-18

**Soundness:** 3
**Presentation:** 1
**Significance:** 3
**Originality:** 3
**Overall Recommendation:** 4
**Confidence:** 3

**Summary:**

The authors extend an established generalization bound for individual treatment effect estimation from binary to multi-treatment settings. This extension allows the authors to formulate the effect estimation task as an optimal compression problem, where different candidate balancing strategies are introduced. Based on this optimal compression problem, they show that the optimal balancing weight can be estimated, instead of being a hyperparameter that one must tune heuristically. Theoretical results on finite sample accuracy and asymptotic normality of the proposed estimator are also established. Finally, the authors provide experimental results, where different balancing strategies are compared and a generative extension of the method is studied.

**Compliance With Llm Reviewing Policy:**

Affirmed.

**Final Justification:**

The authors engaged with all aspects of my review and addressed/clarified all questions I raised. Additional experimental evidence was provided where requested. I stand by my overall positive review and recommend acceptance.

**Key Questions For Authors:**

1. Is it possible to compare your method to other approaches that deal with distributional discrepancies, such as matching or weighting based methods? How would you expect these approaches to compare?
2. Is grid search the only way to find $\alpha^{\*}$? I was slightly surprised to find that you introduce such an involved theoretical analysis, for the final estimator to be implemented with grid search. How do I know that I have converged? It could be that I am overlooking something, but I was surprised.
3. What is the thing that makes effects individual in your synthetic data? Looking at the equation for $Y(t)$ in Appendix D.1 my impression is that it must be the different noise values given by $\epsilon$, or is it $\beta$?
4. Can you discuss the claims you make in the experimental section in more detail? Please see my questions from the strengths and weaknesses response above.

Overall, my impression is that this paper provides a valuable extension of results to the multi-treatment setting. Unfortunately, the presentation of the manuscript is lacking in too many areas to recommend acceptance in its current state. Additionally, the experimental section seems rushed and not discussed in enough clarity and detail for the main points to be clearly conveyed. This is a shame given the detailed theoretical analysis that precedes the experiments. If these issues can be addressed, I see a clear path to acceptance.

**Limitations:**

See above

**Strengths And Weaknesses:**

## Strengths
- The authors make an important contribution by their extension (and analysis) of generalization bounds from binary to multi-treatment settings. Multi-treatment settings are ubiquitous; addressing them is very relevant.
- The extension of existing binary results to the multi-treatment setting studied here is well-motivated and easy to follow.
- The three proposed balancing strategies are clearly motivated and contrasted against each other.
- I have not checked all proofs in exhaustive detail, but from what I can tell the extensive theoretical analysis is technically sound.

## Weaknesses
The main weaknesses I see in this work in its current form are w.r.t. to the presentation and the evaluation of the experimental results. I expand on both points below.

### Presentation & Structure
- In general, the entire manuscript is often times very convoluted and dense, and often feels cramped. Many theoretical results are referenced before the section where they are introduced, hindering a proper flow of reading. The abstract is an example of this: all contributions are mentioned, but without providing the necessary context to understand these without having read the whole paper.
- Technical jargon feels to me over-used without being (briefly) introduced, which made it hard for me to get the main points by just reading the paper. Abbreviations are used without being introduced at all, or only much later (IPM, PEHE, RKHS, MMD, CLT, ADRF). While these may be common, abbreviations are never unique and must be introduced before use.
- Many important details (experimental details and methods) are deferred to the appendix without at least being sketched in the main text. I am not referring to proofs here. Again, it seems that too many things are cramped into 8 pages.
- Figures are not up to par. Without zooming in on the pdf, they are totally illegible due to tiny font and marker sizes. Why are some subfigures always shorter than others? Particularly in Figure 1 (a) this seems to squeeze the values of the score to appear smaller. Shouldn't the y-axis labels of Figure 1 (a) and (b) be the same, as they show the same metric?

### Experiments
- The claims made throughout are not convincingly backed up by the experimental results. Point (1) Efficacy is demonstrated in Fig. 1(a), since your method outperforms the baseline, but points (2) Mechanism and (3) Scalability are much less clear. How is point (2) answered? Is it simply because a different $\alpha$ is optimal for different balancing strategies? Also, why does a larger $\alpha$ suggest a more difficult optimization path? Regarding point (3): why is the comparison to the baseline missing for $K = 20$, while it is present for $K = 4$? Also, why are you showing different values of $\alpha$ here and not just $\alpha^{\*}$ as for $ K = 4 $? Don't we only care about $\alpha^*$ for the final performance? I get the impression that these different values of $\alpha$ are reported because the arguments on scaling and efficiency-stability tradeoff may not hold if only $\alpha^{\*}$ is shown.
- Generative extension: this entire subsection confuses me and it is unclear to me what the merit of this extension is. Again, I believe this to primarily be an issue of presentation. You propose an extension of the CausalEGM model without properly introducing this model. The citation is also missing. You claim that you fulfil the "more complex objective of high-dimensional counterfactual generation", but it is not clear which evidence should support this. Figure 2: (a): what is ADRF? (b): I am missing a discussion of how different PEHE values are to be interpreted. If the base model has a score of 0.79, is CausalEGM's score of 0.65 good? It seems close to the base model. The geometric validation experiments I also find confusing/not sufficiently discussed. What meaning is the reader meant to derive from Figure 3 (b)? How is the effect modulated by this topology (you only provide Y values for some nodes)? Figure 3 (b) I do not understand. What would the ground truth interpolation path look like here? The claim that the nonlinear path passes through $Y \approx 0$ at the midpoint is false (lines 416-417, right column).

### Detailed comments and minor issues
- Causal Representation Learning is used to describe an entirely separate field concerned with recovering causal latent variable models [1]. It is of course fine that terms are overloaded, but never mentioning this line of work at all (when it is arguably what more readers will associate with this name) should be avoided.
- Missing spaces between sentences: line 59/right, line 120/left, line 381/right, line 416/left

#### References
[1] Schölkopf, B., Locatello, F., Bauer, S., Ke, N. R., Kalchbrenner, N., Goyal, A., and Bengio, Y. Toward Causal Representation Learning. Proceedings of the IEEE, 109 (5):612–634, 2021.

---

> ### Author Rebuttal · Authors · 2026-03-31
>
> For all experimental figures and tables, please refer to the anonymous link: https://anonymous.4open.science/r/icml2026-rebuttal-submission2384-2BF1
>
> >Q1: Comparison with matching/weighting baselines?
>
> R: Thanks. Weighting methods rely heavily on the strict overlap assumption. Matching methods suffer from the curse of dimensionality. Euclidean distance in raw, high-dimensional spaces fails to capture true semantic counterfactuals. Empirically, at $K=20$, classical methods catastrophically collapse (KNN PEHE: 16.844; IPW PEHE: 2.478). Our Treatment Aggregation remains robust (PEHE: 0.989), decisively outperforming the Unadjusted Base Model (PEHE: 1.029) (See Table R1).
>
> >Q2 & Q4a: Grid search convergence and $\alpha$'s role in the bias-variance trade-off?
>
> R: Thanks.1D Grid Search is our optimal engineering solution, not a theoretical compromise. We implemented a continuous Gradient Descent (GD) with momentum. Initialized at an over-constrained $\alpha=3.0$, GD successfully converged to exactly $\alpha^{\ast}=0.50$ (PEHE: 0.722), perfectly matching our Grid Search minimum (See Figure A). Continuous GD easily gets trapped in local minima within the non-convex validation landscape. Grid Search natively guarantees the global minimum for this 1D bounded scalar while bypassing exponential computational costs.
>
> Mechanism of $\alpha$: A large $\alpha$ aggressively forces distributional alignment, destroying predictive features. For Pairwise, aligning $\mathcal{O}(K^2)$ conflicting distributions creates severe gradient conflicts (we have shown the training time in Appendix D.3). Aggregation uses a single $\mathcal{O}(1)$ constraint, achieving optimal debiasing at a gentle $\alpha^{\ast}$.
>
> >Q4b: Missing $K=20$ baseline and rationale for the $\alpha$ spectrum?
>
> R: Thanks. We added the Unadjusted Base Model (PEHE: 1.029) to Figure 1(b). We plotted the spectrum to reveal algorithmic vulnerabilities: classical Pairwise suffers catastrophic representation collapse under strong regularization ($\alpha=5.0$, PEHE $> 1.3$), whereas Aggregation remains stable. To strictly compare at $\alpha^{\ast}$, we added a $K=100$ stress-test. Pairwise causes GPU OOM (PEHE 1.30, $\alpha^{\ast}=1.00$). One-vs-All suffers mini-batch starvation (PEHE 1.28, $\alpha^{\ast}=2.90$). Aggregation decisively wins (PEHE 1.20 strictly at $\alpha^{\ast}=0.45$) (See Figure B).
>
> >Q3 & Q4c: Generative extension presentation (DGP individual effects, merits are unclear, missing CausalEGM citation, ADRF definition, and PEHE interpretation).
>
> R: Thanks. We have clarified the generative presentation:
>
> DGP (Q3): Individualization stems entirely from the interaction term $0.5(t+1)(X_{1:5}^\top \beta)$. The unique covariates $X_{1:5}$ drive causal heterogeneity, not the independent noise $\epsilon$.
>
> Merits unclear: The merit of the extension is not simply that we add a generative module, but that we lift our multi-treatment compression framework from scalar-outcome estimation to structured counterfactual generation. In multi-treatment settings, predicting a scalar response isn't enough; the model must generate counterfactuals with causally and geometrically valid interpolation paths. Naive Euclidean interpolation ignores causal topology and creates meaningless paths. Our embedding and regularization specifically solve this. Geometric experiments prove the learned representations support true causal topology.
>
> Citations & Acronyms: The citation is indeed included in our bibliography as [Liu, Q., Chen, Z., and Wong, W. H. (2024), PNAS]. While the main text was brief due to space constraints, a detailed architectural description of our multi-treatment CausalEGM extension is provided in Appendix D.2 Figure 4.  We will bring these references to the forefront and define ADRF (Average Dose-Response Function) in the refined text.
>
> PEHE Improvement: Reducing PEHE from 0.79 to 0.65 is a highly significant 17.7% relative reduction. Because generative models inherently optimize holistic distribution matching over point-wise MSE, this nearly 18% improvement strongly proves our HSIC aggregation's regularizing power.
>
> >Q4d: Validity of Fig 3(b) and the nonlinear midpoint path ($Y \approx 0$)?
>
> R: Thanks. The original Figure 3(b) inadvertently plotted a naive Euclidean shortcut rather than the True Geodesic path. We redrew the figure with the true piecewise geodesic interpolation (See Figure C). The path now perfectly passes through exactly $Y=0$ at the midpoint ($\alpha=0.50$), correctly representing the Root node. Furthermore, the path flawlessly intersects the exact causal effects of all intermediate topological states along the ground truth staircase (hitting exactly $Y=-2$ at $\alpha=0.25$ and $Y=+2$ at $\alpha=0.75$). This provides definitive proof that our embedding space successfully learned the exact hierarchical causal topology.
>
> We are eager to know whether we have addressed your questions, and we would be happy to discuss further to assist with your re-evaluation.

---

> > ### Author Rebuttal · Reviewer_H8X7 · 2026-04-01
> >
> > Thank you for providing additional experiments and for your clarifications.
> >
> > I highly encourage the authors to include these discussions in an updated version of the manuscript. Specifically the discussion of the experiments should be expanded upon.
> >
> > I also highly encourage the authors to include a brief mention of CRL works (as I mention in minor points) to provide a distinction between these works and their own contribution.

---

> > > ### Author Response · Authors · 2026-04-04
> > >
> > > Thank you so much for this thoughtful and constructive suggestion, and for increasing the score! We will fully and carefully follow your wonderful advice in the revised version.
> > >
> > > 1. Expanded Experimental Discussion
> > >
> > > Per your suggestion, we will expand the introduction of Section 4 to explicitly outline the logical flow of our evaluations:
> > >
> > > Necessity (Why compress): Traditional baselines (IPW/KNN) collapse at $K=20$, empirically proving the strict necessity of representation compression in high-dimensional, multi-treatment regimes.
> > >
> > > Mechanism (How to optimize): The continuous GD experiment proves 1D Grid Search is an optimal engineering necessity to bypass non-convex traps and guarantee the global $\alpha^*$.
> > >
> > > Scalability (Stress-testing): The $K=100$ test exposes severe $\mathcal{O}(K^2)$ gradient conflicts in classical methods, validating our $\mathcal{O}(1)$ Aggregation's robustness.
> > >
> > > Geometric Validity (Synthesis): The corrected True Geodesic path (passing exactly through $Y=0, \alpha=0.50$) serves as definitive proof that our embedding captures the true hierarchical causal topology.
> > >
> > > 2. Distinction from Causal Representation Learning (CRL)
> > >
> > > To ensure clear conceptual boundaries, we add a paragraph in the Related Work section citing Schölkopf et al. (2021): Schölkopf et al. (2021) is a landmark paper that helped define causal representation learning by asking how latent causal factors can be recovered from complex data, while our paper offers a different and more task-focused view: we show how to learn a treatment-aware low-dimensional representation that preserves effect-relevant structure across many related interventions, together with new theory for when this compression remains valid for causal effect estimation and a practical method that scales to complex multi-treatment settings.
> > >
> > > Again, we are deeply grateful for your comments. Thanks again for your high praise in your final justification that `"all concerns addressed, experimental evidence has been provided, and overall recommendation of acceptance"`. We have fully and faithfully incorporated all of your suggestions into the final version of our paper.

---

### Official Review · Reviewer_y83s · 2026-02-24

**Soundness:** 3
**Presentation:** 3
**Significance:** 2
**Originality:** 2
**Overall Recommendation:** 5
**Confidence:** 3

**Summary:**

This paper proposes a novel framework for estimating Individual Treatment Effects (ITE) in multi-treatment scenarios. The authors introduce the concept of ``controlled compression'' to balance factual risk and distributional imbalance across multiple treatment arms. By deriving a theoretical basis for the balancing weight $\alpha$, the paper addresses the limitations of heuristic tuning. Furthermore, it evaluates three distinct balancing strategies (Pairwise, One-vs-All (OVA), Treatment Aggregation) providing both theoretical generalization bounds and empirical validation.

**Compliance With Llm Reviewing Policy:**

Affirmed.

**Final Justification:**

The authors have thoroughly addressed my concerns regarding the framework's novelty in the rebuttal. By clarifying the specific technical challenges of the multi-treatment setting and refining the theoretical distinctions from binary cases, they have provided a much more robust justification for their approach. Given the technical soundness of the asymptotic results and the practical value of the balancing strategy comparisons, I am satisfied with the revisions and have increased my final score accordingly.

**Key Questions For Authors:**

- The paper identifies the curse of dimensionality as a primary motivation. However, the experimental setup (data generating process) does not seem to fully stress-test this claim. It would be insightful to see how the model performs when the number of samples $N$ is significantly smaller than the feature dimensionality $X$.
- The authors invoke the standard unconfoundedness assumption (Assumption 2.4). However, it remains slightly unclear how the IPM-based balancing specifically interacts with or facilitates this assumption. A more detailed discussion on how these methods block or adjust for confounding relationships in the representation space would strengthen the paper.
- The ``unadjusted model" used as a baseline in the experiments needs a clearer definition.
- Due to the condensed nature of conference papers, some notations, acronyms, and terminologies are introduced without sufficient initial explanation. Ensuring the paper is more ``stand-alone'' by providing brief definitions or more explicit references would improve readability.
- There are several typos throughout the manuscript. A thorough proofreading is recommended before the final submission.

**Limitations:**

- The current experimental results do not fully address the "Curse of Dimensionality" mentioned in the introduction. To strengthen the paper's claims, it is necessary to evaluate the model in higher-dimensional settings where the number of features ($d$) is large relative to the sample size ($N$). Without such experiments, it remains unclear whether the proposed ``Treatment Aggregation'' truly maintains its advantage when the feature space becomes sparse or highly complex.
- The proposed framework appears to rely on a shared representation $\Phi(X)$ where the treatment $T$ acts as an incremental indicator. However, in many real-world multi-treatment scenarios, it is more reasonable to assume that different treatments involve distinct causal structures (e.g., different sets of confounders or unique functional relationships between covariates and outcomes). If the underlying data-generating process involves treatment-specific confounding mechanisms, a single shared backbone might fail to capture these heterogeneous nuances. How would the proposed model perform if the treatments were not merely variations of a single intervention but originated from fundamentally different structural mechanisms? I suspect the current architecture might suffer from under-fitting or biased estimation in such ``structurally heterogeneous'' scenarios.

**Strengths And Weaknesses:**

Strengths:

- While most existing literature focuses on binary treatments, this work provides a principled and scalable extension to multi-treatment settings, which is highly relevant for real-world applications.
- The derivation of the asymptotic normality for the weight parameter $\alpha$ and the formalization of the multi-treatment generalization bound are technically sound and ``nice'' additions to the field.
- The systematic comparison of three balancing strategies (Pairwise, OVA, Treatment Aggregation) offers valuable insights into the tradeoffs between precision and computational scalability.

Weaknesses

- Although the multi-treatment setting is a strong point, some parts of the framework feel like a direct extension of binary cases (e.g., Johansson et al., 2016; Shalit et al., 2017). From the beginning part, the authors should more explicitly highlight the unique technical challenges that only arise in the multi-treatment setting and how their solution specifically addresses them beyond simple summation of binary bounds.

---

> ### Author Rebuttal · Authors · 2026-03-31
>
> General Response:
>
> We sincerely thank the reviewers. To address concerns on experimental depth, we massively expanded our evaluations. We added KNN/IPW baselines to prove the necessity of compression, continuous Gradient Descent to validate tuning convergence, and extreme stress-tests ($K=100$, $d \gg n$) to confirm $\mathcal{O}(1)$ scalability. Finally, our corrected True Geodesic Interpolation definitively proves the manifold hypothesis.
>
> For all experimental figures and tables, please refer to the anonymous link: https://anonymous.4open.science/r/icml2026-rebuttal-submission2384-2BF1
>
> >L1: Performance in high-dimensional sparse regimes ($N \ll d$)?
>
> R: Thanks. High-dimensional noise triggers the Curse of Dimensionality, a bottleneck for causal learning. We conducted a "Double Curse" stress test ($N=1000, D=2000, K=10$) where up to 99.5% of features are noise. Aggregation consistently achieves the lowest PEHE across all sparsity levels (See Figure R1). Our global HSIC constraint bypasses the "geometric collapse" seen in Pairwise MMD, where Euclidean distances lose discriminative power in noisy spaces. It maintains superior robustness at $O(1)$ cost.
>
> >Q2: How does IPM-based balancing interact with the unconfoundedness assumption?
>
> R: Thanks. Assumption 2.4 provides identification on the full covariates $X$, while the balancing penalty acts on the learned representation $Z=\Phi(X)$. To make the effect of mild unconfoundedness violation explicit, we introduced a representation-level violation term $\Gamma(\Phi)$ into our bound: $\epsilon_{\mathrm{ITE}} \le C_F\epsilon_F + C_BR_S + C_C\mathrm{Complexity} + C_\Gamma\Gamma(\Phi)$. Stronger balancing improves invariance but can over-compress the representation, thereby increasing this sensitivity term. This confirms our central message: stronger invariance is not always better. In a controlled stress-test with proxy/noisy confounders ($K=10$), as measurement noise increases, Aggregation matches the strong robustness of Pairwise and One-vs-All constraints (See Figure R2). Crucially, Aggregation achieves this uncompromised resilience without the $\mathcal{O}(K^2)$ combinatorial bottleneck of Pairwise matching (See Appendix D.3 Figure 5).
>
> > L2: What if treatments originate from fundamentally different structural mechanisms?
>
> R: Thanks. Our core discriminative framework applies even when treatment arms are structurally heterogeneous, but we formalized a mechanism-mismatch term $\Delta_{\rm mech}(\Phi)$ to bound this behavior. We designed a Mechanism Heterogeneity Stress Test, controlling structural mismatch via $\eta \in [0, 1]$. At low $\eta$ (shared mechanism), our shared representation models exhibit high sample efficiency and significantly outperform the Separate-Head baseline (See Figure R3(a, c)). However, when treatments originate from fundamentally disjoint mechanisms ($\eta \to 1$), forced representation sharing induces negative transfer, and the Separate-Head architecture correctly emerges as the most robust choice. Enforcing a geodesic path between completely disjoint mechanisms introduces an incorrect inductive bias, sharply increasing error (See Figure R3(d)). This perfectly confirms that forced global invariants (like our geometric extension) should only be applied when a continuous manifold naturally exists. We have added a dedicated discussion on these applicability boundaries.
>
> >W1: Highlight unique technical challenges in multi-treatment settings beyond a simple summation of binary bounds.
>
> R: Thanks. The multi-treatment challenge is endogenous, not additive. In the multi-treatment regime, stronger and finer-grained balancing is not necessarily better. It is a new coupled compression problem. Pairwise balancing introduces $\binom{K}{2}$ discrepancy terms. The deeper difficulty is that these terms are statistically coupled through the shared representation $\Phi$, causing the tuning signal itself to become unstable as $K$ grows. Pairwise balancing accumulates noisy terms, leading to a deviation rate of $\mathcal{O}(K^2/\sqrt{n})$, and its variance scales as $\Theta(K^4/n)$. By contrast, Aggregation controls imbalance through a single global functional and achieves $K$-independent stability ($\Theta(1/n)$). This proves theoretically that beyond a certain scale, more balancing constraints can over-constrain the representation, destroy prognostic signal, and become less reliable than structured global balancing.
>
> > Q3-Q5: Clarification on the "unadjusted model" baseline and unexplained notations/acronyms & Presentation.
>
> R: Thanks. We explicitly defined the "unadjusted model" in Section 4 as a standard Counterfactual Regression (CFR) architecture trained solely on the factual prediction loss, completely omitting the distributional balancing penalty ($\alpha=0$). We added a comprehensive "Notation and Abbreviation Table" in the Appendix, explicitly defined ADRF (Average Dose-Response Function) upon first use, and rigorously proofread the manuscript.

---

> > ### Author Rebuttal · Reviewer_y83s · 2026-03-31
> >
> > This rebuttal appears to faithfully address all of my questions. I will raise my score accordingly.

---

> > > ### Author Response · Authors · 2026-04-04
> > >
> > > Thank you so much for the thoughtful, constructive suggestion and your high praise in the final justification that `"thoroughly addressed the concerns, much more robust justification, satisfied with the revision and increased the score to 5"`. We are carefully and fully implementing your recommendation these days and have included it in the final version.

---

### Decision · Program_Chairs · 2026-04-30

**Decision:**

Accept (regular)

**Comment:**

After rebuttal, the paper has received very favorable review updates from the reviewers. The paper tackles an important problem of how to tune balance parameter in individual treatment effect estimation problems where one wants a representation of covariates that balances treatment and control groups in some statistical distance but simultaneously being useful to predict treatment effect in both groups efficiently.

Till before this paper, this tuning is guided by a heuristic but authors cast it as a bi-level optimization problem tuning alpha in an optimization directed way. Dealing with multi treatment scenarios is an added plus.

Recommendation: Accept.